# TMD-Bench: A Multi-Level Evaluation Paradigm for Music–Dance Co-Generation

Xiaoda Yang [* 1]  Majun Zhang [* 1]  Changhao Pan [1]  Nick Huang [2]  Yuguang Yang [2]  Fan Zhuo [1]  Pengfei Zhou [3]
Jin Zhou [2]  Sizhe Shan [1]  Shan Yang [2]  Miles Yang [2]  Yang You [3]  Zhou Zhao [1]

## Abstract

Unified audio–visual generation is rapidly gaining industrial and creative relevance, enabling applications in virtual production and interactive media. However, when moving from general audio–video synthesis to music–dance co-generation, the task becomes substantially harder: musical rhythm, phrasing, and accents must drive choreographic motion at fine temporal resolution, and such rhythmic coupling is not captured by unimodal metrics or generic audiovisual consistency scores used in current evaluation practice. We introduce **TMD-Bench**, a benchmark for text-driven music–dance co-generation that assesses systems across unimodal generation quality, instruction adherence, and cross-modal rhythmic alignment. The benchmark integrates computable physical metrics with perceptual multimodal judgments, and is supported by a curated rhythm-aligned music–dance dataset and a fine-grained Music Captioner for structured music semantics. TMD-Bench further reveals that (i) modern commercial audio–visual models (e.g., Veo 3, Sora 2) produce high-quality music and video, while rhythmic coupling remains less consistently optimized and leaves room for improvement, and (ii) our unified baseline **RhyJAM** trained on rhythm-aligned data achieves competitive beat-level synchronization while maintaining competitive unimodal fidelity. This presents prospects for building next-generation music–dance models that explicitly optimize rhythmic and kinetic coherence. The code is available at https://github.com/MM-Speech/TMD-Bench.

*Equal contribution  [1]Zhejiang University  [2]Tecent, China  [3]School of National University of Singapore.  Correspondence to: Zhou Zhao <zhaozhou@zju.edu.cn>.

*Proceedings of the $43^{rd}$ International Conference on Machine Learning*, Seoul, South Korea. PMLR 306, 2026. Copyright 2026 by the author(s).

## 1. Introduction

Audio–visual synthesis is gaining increasing industrial and creative relevance, enabling applications in filmmaking (Polyak et al., 2024; Zhang et al., 2025b) and interactive media (Hoi et al., 2025; Jiang et al., 2025; Wang et al., 2025c). Rather than generating sound and imagery in isolation, audio–visual synthesis seeks to produce outputs that are both temporally and semantically coherent across modalities. Recent research efforts have further advanced this direction, including commercial systems such as Veo 3 (DeepMind, 2025) and Sora 2 (OpenAI, 2025), and open-source counterparts such as Ovi (Low et al., 2025), JavisDiT (Liu et al., 2025), and LTX-2 (HaCohen et al., 2026).

Despite the rapid progress in audio–visual generation, existing models often struggle when applied to music–dance co-generation. Unlike generic audio–visual settings that primarily require cross-modal semantic correspondence and coarse temporal synchrony, music–dance generation demands more rigorous coupling between modalities. In particular, musical rhythm, phrasing, and expressive accents impose fine-grained temporal structure that must be reflected in full-body motion through beat-level alignment, coordinated limb trajectories, and biomechanically plausible transitions.

While recent systems have begun to explore music–dance co-generation, the corresponding evaluation methodology has not kept pace. Current evaluation practice for audio–visual generation (e.g., VABench (Hua et al., 2025)) is largely grounded in unimodal assessments of audio or video quality and generic audiovisual consistency measures that evaluate semantic correspondence, temporal co-occurrence, and conformity to scene-level physical and logical constraints. Such metrics are suitable for general audio–video synthesis—e.g., ensuring that sound effects match visual events or that narrated content corresponds to depicted scenes—but they fail to capture the intrinsic coupling and fine-grained alignment between musical rhythm and choreographic kinetics required in music–dance generation.

To bridge this gap, we introduce a comprehensive benchmark for text-driven music–dance co-generation that evaluates audio and motion as intrinsically coupled generative

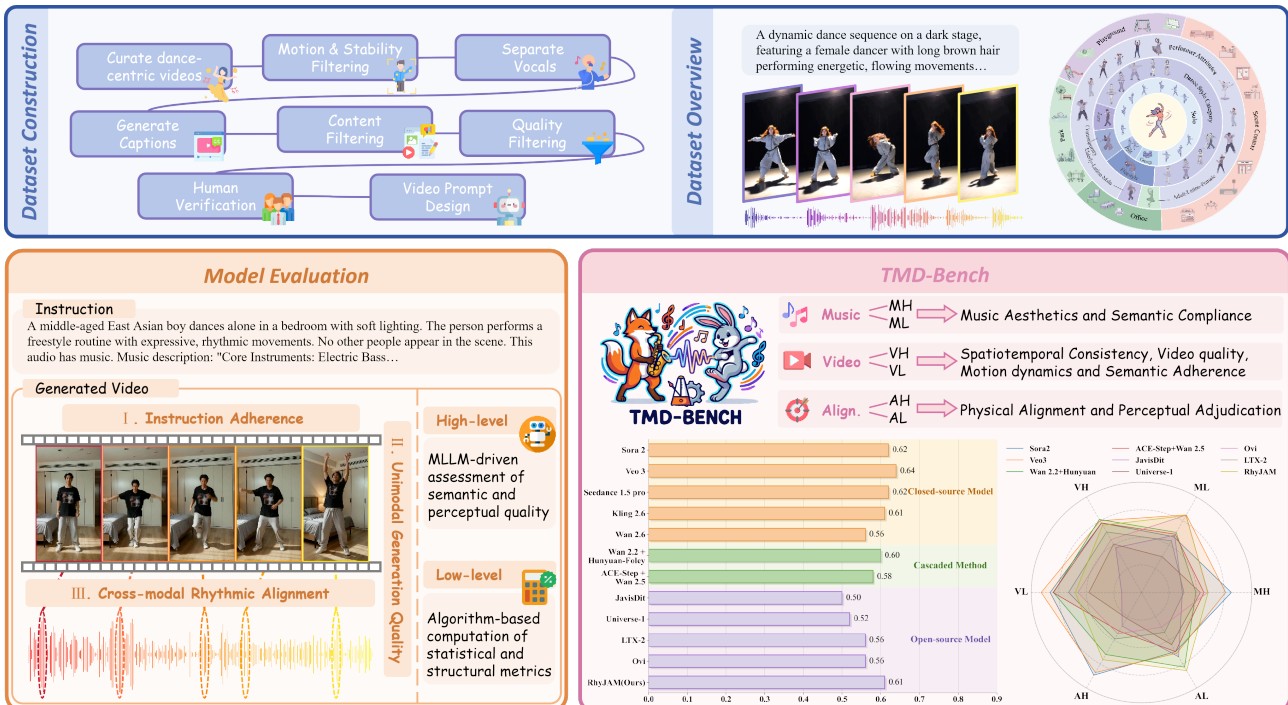

*Figure 1.* **Overview of TMD-Bench, a benchmark for text-driven music–dance co-generation.** The top-left panel illustrates the dataset construction pipeline. The dataset overview (top-right) shows the distribution of dance-related attributes in the 10k dataset; each concentric ring corresponds to an attribute—Performer Cardinality, Dance Style Category, Performer Attributes, and Scene Context (from inner to outer). The evaluation protocol (bottom-left) assesses models across (I) instruction adherence, (II) unimodal generation quality, and (III) cross-modal rhythmic alignment. The bottom-right summarizes the TMD-Bench metric taxonomy covering music, video, and alignment, along with comparative performance across closed-source, cascaded, and open-source models.

dimensions. Our framework broadly assesses systems along three complementary axes: (i) unimodal generation quality for both music and dance, (ii) unimodal instruction adherence with respect to textual specifications, and (iii) cross-modal rhythmic and kinetic alignment between modalities. To operationalize these criteria, we develop a two-level evaluation pipeline comprising both low-level physical metrics and high-level perceptual assessments. The physical layer computes signal- and event-based statistics to quantify rhythmic structure, kinetic dynamics, and alignment patterns from a measurable standpoint, while the perceptual layer leverages multimodal large language models (MLLM) to approximate human judgments regarding semantic coherence, stylistic intent, and cross-modal synchronization. This design notably enables systematic comparison, analysis, and development of music–dance co-generation models within a unified benchmarking framework in practice.

Beyond providing a unified evaluation protocol, TMD-Bench also reveals several systematic insights. First, commercial audio–visual generators such as Veo 3, Sora 2, and Kling 2.6 (Kuaishou, 2025) attain strong unimodal performance in both quality and instruction adherence, yet their music–dance rhythmic alignment is generally competitive and often comparable to our unified model **RhyJAM** (**Rhy**thm **J**oint **A**udio–**M**otion Genera-

tion Model), suggesting that beat-level coupling remains a shared challenge and leaves room for further improvement even in strong general-purpose systems. Second, existing open-source audio–video models remain weaker overall, while cascaded text-to-dance-to-music and text-to-music-to-dance pipelines achieve somewhat better—but still incomplete—performance; in contrast, **RhyJAM**, trained on a high-quality music–dance corpus, markedly improves rhythmic synchronization to competitive levels while preserving competitive unimodal generation, suggesting an inherent advantage of unified architectures for cross-modal consistency. Finally, a clear gap persists between closed-source and open-source models across dimensions, underscoring the need for continued community efforts toward strong, openly available music–dance co-generation systems.

Our contributions are three-fold:

- **TMD-Bench.** We propose the first benchmark tailored to music–dance co-generation, featuring a comprehensive multi-level evaluation framework that integrates unimodal quality assessment with cross-modal rhythmic alignment analysis from both physical and perceptual perspectives.

- **Unified Generation Model (RhyJAM).** We develop a unified text-to-music–dance diffusion model that

jointly generates music and choreographic motion under shared semantic conditioning, providing a strong open-source baseline within our evaluation framework.

- **Rhythm-Aligned Suite.** We curate a 10k-scale rhythm-aligned music–dance dataset together with a specialized Music Captioner for fine-grained music semantics, supporting both training and standardized evaluation.

## 2. Related Work

### 2.1. Music-Driven Human Motion Synthesis

The synthesis of human motion from music has evolved from generating coarse 3D skeletons to producing expressive, full-body performances. Early methods often treated dance generation as a sequence modeling task (Lee et al., 2019). To improve the expressiveness of generated dance, Li et al. (2023) proposed FineNet, which utilizes a diffusion-based network to model fine-grained hand and body movements. Recent advancements have shifted towards direct video synthesis. For instance, X-Dancer (Wang et al., 2025a) employs a unified transformer-diffusion framework for zero-shot music-to-dance generation, leveraging an autoregressive transformer to produce 2D pose sequences that guide a diffusion model for high-fidelity video synthesis. Furthermore, maintaining long-term synchronization remains a challenge. MoMu-Diffusion (You et al., 2024) addresses this by proposing a Bidirectional Contrastive Rhythmic VAE to extract modality-aligned latent representations, ensuring that the generated motion remains temporally consistent with the musical rhythm over extended durations.

### 2.2. Unified Audio-Video Joint Generation

Traditionally, audio–video generation relied on cascaded pipelines, such as video-to-audio (Cheng et al., 2024; Liang et al., 2025) or audio-to-video (Adi et al., 2023) synthesis. However, these decoupled approaches often suffer from poor semantic alignment and temporal jitter. Recent research has pivoted towards unified architectures that model both modalities within a single generative process. The foundation for high-quality audio in these frameworks has been bolstered by models like ACE-Step (Gong et al., 2025), which utilizes a flow-matching-based linear transformer for rapid, high-fidelity music and singing synthesis.

On the architectural side, several unified Diffusion Transformer variants have emerged. Recent surveys on text-to-video generation summarize the rapid progress of diffusion- and transformer-based video synthesis, while also highlighting persistent challenges in temporal consistency and controllability (Xie et al., 2025). JavisDiT (Liu et al., 2025) introduces a Hierarchical Spatio-Temporal Prior Synchronization mechanism to align visual regions with auditory frequencies, while Ovi (Low et al., 2025) adopts a twin-backbone fusion strategy using blockwise cross-attention and scaled-ROPE embeddings to achieve natural synchronization. UniAVGen (Zhang et al., 2025a) incorporates asymmetric cross-modal interactions and Face-Aware Modulation to improve lip synchronization and facial expressivity, whereas UniVerse-1 (Wang et al., 2025b) exemplifies an expert-stitching paradigm that fuses specialized video and music experts into a unified model. More recently, LTX-2 (HaCohen et al., 2026) extends unified text-to-audio-video generation with an efficient asymmetric dual-stream design and bidirectional audio-video cross-attention.

Prior multimodal works have explored lip-readable synchronization, audio-augmented recognition, camera-controllable generation, controllable retrieval, and egocentric reasoning (Yang et al., 2024b;a; 2025b;a; 2026a;b), but they do not directly evaluate fine-grained rhythmic coupling in text-driven music–dance co-generation. Compared to traditional audio–visual consistency tasks (e.g., speech–lip synchronization or collision–sound correspondence), music–dance coherence hinges on fine-grained coupling between the rhythmic structure of a largely continuous music stream and full-body motion accents. Recent modality-aware reward modeling, such as SDiaReward (Lu et al., 2026), also highlights the need to evaluate modality-specific perceptual cues beyond text-level semantics. Thus, even if unified audio–video models achieve strong semantic consistency and coarse synchrony, outputs can still look plausible yet feel off-beat, with looseness or drift. To systematically capture and measure this finer-grained rhythmic alignment challenge, we introduce TMD-Bench, a unified protocol that evaluates instruction adherence, unimodal quality, and cross-modal rhythmic coherence.

## 3. TMD-Bench

In text-driven music–dance joint generation, the primary challenge of evaluation arises from a fundamental mismatch between existing audio–visual evaluation paradigms and the nature of the task itself. In dance-centric scenarios, human judgments emerge from a multi-dimensional and simultaneous evaluation process that integrates perceptual content quality, semantic alignment with textual instructions, and coherent rhythmic coupling between music and motion. Such judgments are inherently open-ended and admit multiple valid realizations, rendering pointwise alignment with a single ground-truth instance both unfair and insufficient. Moreover, purely algorithmic metrics that operate on isolated signals or event matches cannot fully capture the perceptual qualities that matter in practice; high-level, intuition-aligned perceptual evaluation plays an equally critical role in assessing the effectiveness of music–dance generation.

Motivated by these considerations, we organize the evaluation of joint generation into a triadic framework that

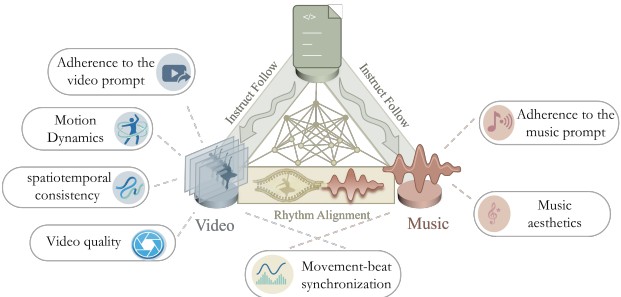

*Figure 2.* Overview of our evaluation framework for music–dance generation. The benchmark decomposes video, audio, and cross-modal alignment into complementary dimensions.

encompasses generation quality and semantic adherence and audio–visual rhythmic alignment, as Figure 2 depicts. Crucially, we adopt a layered paradigm that combines computable low-level metrics with high-level adjudication endowed with semantic and perceptual reasoning capabilities. The lower layer emphasizes stability and scalability through quantitative measures, whereas the higher layer leverages an MLLM-as-a-Judge mechanism to capture compositional semantics and rhythm perception that resist reduction to individual scalar metrics.

### 3.1. Audio Evaluation

For the audio modality, evaluation comprises two orthogonal criteria: music aesthetics and semantic compliance, characterizing the quality and faithfulness of the audio.

At the level of audio quality, we adopt a set of dimensions that have been widely used in prior work (Tjandra et al., 2025) and shown to be stable in practice: **Production Quality (PQ)**, **Production Complexity (PC)**, **Content Enjoyment (CE)**, and **Content Usefulness (CU)**. To further address aspects that are difficult to fully express through algorithmic measures, we additionally introduce a MLLM-based MOS simulation. By approximating human subjective judgments at a higher level of abstraction, this component serves as an orthogonal perceptual evaluation axis that balances quantitative stability with subjective consistency.

For instruction adherence, we first compute **CLAP** (Wu et al., 2024) cosine similarity between the text prompt and the generated audio. Since music is inherently multi-attribute (e.g., instrumentation, rhythm and genre), a single similarity score is often insufficient. We therefore introduce an open-source Music Captioner fine-tuned from Qwen-Omni (Xu et al., 2025), which produces aligned captions over six dimensions (core instruments, rhythm & groove, tempo, genre, ambiance & emotion, and functional scenes; see Figure 3). During assessment, we perform dimension-wise comparisons between the outcome of captioner and the prompt-specified semantics, enabling fine-grained and interpretable verification of semantic compliance. Compared

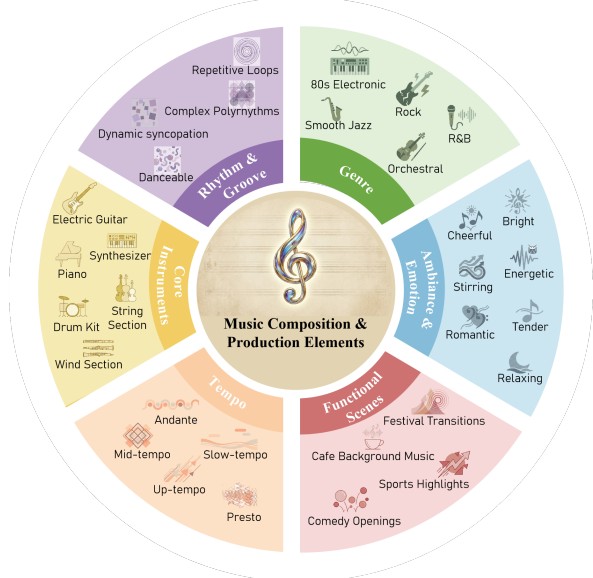

*Figure 3.* Overview of the six semantic dimensions used by the Music Captioner for structured audio annotation.

with closed-source LLM adjudication, our captioner enables stable, consistent, and scalable automatic annotation for systematic benchmark construction.

### 3.2. Video Evaluation

We anchor video evaluation along three intrinsic dimensions—spatiotemporal consistency, video quality, and motion dynamics—and further assess semantic adherence to the textual prompt.

At the algorithmic level, we adopt a set of computable VBench-style (Huang et al., 2023) metrics. Specifically, **Subject Consistency** and **Background Consistency** measure the temporal stability of the subject identity and scene structure; **Imaging Quality** and **Aesthetic Quality** assess visual quality from the perspectives of technical imaging fidelity and holistic visual appeal; **Dynamic Degree** quantifies the magnitude of actual motion displacement and **Motion Smoothness** evaluates temporal continuity and the smoothness of motion trajectories. In addition, we incorporate **ViCLIP** (Wang et al., 2024) metric, which computes cross-modal similarity between the video content and textual instructions to quantify semantic compliance.

Moreover, we further introduce the VLM-as-a-Judge as a complementary perceptual and reasoning layer. **Instruction Following** evaluates semantic instruction adherence, deriving scores by contrasting observed video content with the instruction-implied ideal description. **Quality** delivers an integrated assessment of imaging fidelity and aesthetic completeness from a high-level perceptual perspective. **Motion** and **Consistency** assess motion continuity, movement amplitude, and spatial stability from a holistic viewpoint,

acting as perceptual counterparts to algorithmic motion- and consistency-based measures.

### 3.3. Audio–Visual Alignment Evaluation

We next focus on the central objective of joint generation: temporal alignment between musical rhythmic structure and dance motion accents. Purely subjective judgments are neither reproducible nor scalable, whereas relying on physical time matching fails to capture perceptual deviations arising from nonlinear rhythmic variation, hierarchical energy dynamics, and stylistic differences in motion execution.

To address this limitation, we introduce **MDAlign**, a dual-track evaluation framework that integrates physical alignment metrics with perceptual adjudication. In this design, the physical track provides computable and scalable alignment measures, while the perceptual track employs a reasoning-capable MLLM to approximate human holistic judgments of rhythmic coherence.

At the physical alignment level, MDAlign maps audio and video into discrete event representations on a shared temporal axis. For the audio modality, we extract a set of beat timestamps $A = \{a_1, a_2, \ldots, a_n\}$, where $a_i$ denotes the temporal position of the $i$-th musical beat. For the video modality, we first apply a pose estimation model to extract the positions of $K$ keypoints in each frame, denoted as $P_t \in \mathbb{R}^{K \times 2}$. We then compute the average inter-frame displacement of keypoints to obtain a motion velocity signal:

$$V(t) = \frac{1}{K} \sum_{k=1}^{K} \left\| P_{t+1}^{(k)} - P_t^{(k)} \right\|_2 \qquad (1)$$

To suppress high-frequency jitter, this signal is further smoothed using a one-dimensional Gaussian kernel. The temporal locations of its local maxima are defined as the motion accent set $M = \{m_1, m_2, \ldots, m_{|M|}\}$, which represents salient changes in dance motion along the time axis.

Based on these representations, we define two physical alignment metrics. The first is the *Video Beat Consistency Score* (VBCS), which measures the temporal proximity between each motion accent and its nearest musical beat:

$$\text{VBCS} = \frac{1}{|M|} \sum_{m \in M} \exp\left( -\frac{\min_{a \in A} |m - a|^2}{2\sigma^2} \right) \qquad (2)$$

where $\sigma$ is a temporal smoothing hyperparameter that controls tolerance to small timing deviations. This metric emphasizes whether motion accents occur near musical beats and is robust to minor temporal offsets.

The second metric is the *Audio Beat Hit Score* (ABHS), which evaluates alignment from a coverage perspective by measuring the degree to which musical beats are effectively and reliably responded to by motion in practice:

$$\text{ABHS} = \frac{1}{|A|} \sum_{a \in A} \mathbb{I}\left( \min_{m \in M} |a - m| < \tau \right) \qquad (3)$$

where $\tau$ denotes a temporal hit window and $\mathbb{I}(\cdot)$ is the indicator function. While VBCS focuses on temporal precision, ABHS quantifies the mismatch between musical beats and the corresponding motion responses.

Unlike paired ground-truth beat alignment metrics (e.g., BHS and BCS in MoMu-Diffusion (You et al., 2024)), VBCS and ABHS operate purely on rhythmic events extracted from the generated audio–video output, making them suitable for open-ended joint generation scenarios. To balance the complementary objectives of avoiding spurious beats and missing beats, we define the overall physical alignment score as the arithmetic mean. Additionally, we report CSD and HSD as stability measures, defined as the empirical variances of VBCS and ABHS, respectively, where lower values indicate more consistent alignment behavior.

Nevertheless, physical event matching alone is insufficient to capture perceptual rhythmic coherence. When musical energy intensifies or motion amplitude evolves across segments, pointwise alignment may still appear correct while the rhythm feels loose or lacks tension. To address this gap, MDAlign introduces an MLLM-based perceptual alignment adjudication that focuses on the correspondence between audio pulses and motion accents, complementing physical matching with higher-level understanding of complex rhythmic structure and expressive motion.

## 4. RhyJAM: Unified Model for Music–Dance Generation

### 4.1. Flow-Matching Formulation

We adopt flow matching to learn a continuous-time transport in the latent space, parameterized by a conditional velocity field $v_\theta(\cdot, t\,;\,c)$. Let $\rho_t(z)$ denote the density of latent variable $z$ at time $t \in [0, 1]$. The probability flow satisfies the continuity equation

$$\partial_t \rho_t(z) \;+\; \nabla \cdot \big( \rho_t(z)\, v_\theta(z, t\,;\,c) \big) \;=\; 0. \qquad (4)$$

Sampling is performed by integrating the neural ODE

$$\frac{dz(t)}{dt} = v_\theta\big(z(t), t\,;\,c\big). \qquad (5)$$

In our convention, the trajectory starts from Gaussian noise and is integrated backward to the data manifold, i.e., $z(1) \sim \mathcal{N}(0, I)$ and $z(0) \sim p_{\text{data}}$.

To obtain a simulation-free training objective consistent with our unified schedule, we define a tractable probability

path by mixing clean latents with noise:

$$z_t = \alpha(t)\, z_0 + \sigma(t)\, \varepsilon,$$

$$v^*(t, z_0, \varepsilon) := \frac{dz_t}{dt} = \dot{\alpha}(t)\, z_0 + \dot{\sigma}(t)\, \varepsilon, \tag{6}$$

where $z_0 \sim p_{\text{data}}$ and $\varepsilon \sim \mathcal{N}(0, I)$ are sampled once per training instance, and $\alpha(t), \sigma(t)$ are smooth functions. The flow matching loss regresses the model velocity to the analytic target along this path:

$$\mathcal{L}_{\text{FM}} = \mathbb{E}_{t \sim \mathcal{U}[0,1],\, z_0,\, \varepsilon} \left[ \left\| v_\theta(z_t, t\,;\, c)\ -\ v^*(t, z_0, \varepsilon) \right\|_2^2 \right]. \tag{7}$$

### 4.2. Model Architecture

As illustrated in Figure 4, we adopt an end-to-end architecture that models audio and video within a single diffusion process, enabling text-driven music–dance co-generation.

Given audio $x^a$, video $x^v$, and text $c$, the model encodes the two modalities into latent variables $z_0^a = \mathcal{E}_a(x^a)$ and $z_0^v = \mathcal{E}_v(x^v)$, while the textual input produces a conditioning representation $h_c$ that provides shared semantic guidance.

To explicitly align the two modalities along the diffusion timeline, we apply noise perturbations to both audio and video latent variables at the same diffusion time step $t$. Using a unified noise schedule $\alpha(t), \sigma(t)$, the noisy latent variables are constructed as

$$z_t^m = \alpha(t)\, z_0^m + \sigma(t)\, \varepsilon^m \qquad m \in \{a, v\} \tag{8}$$

where $\varepsilon^a, \varepsilon^v \sim \mathcal{N}(0, I)$ are independently sampled Gaussian noise variables.

The fusion module updates latents through a sequence of attention operations that progressively incorporate temporal structure, textual semantics, and cross-modal correspondence. Letting $m \in \{a, v\}$ denotes the modality and $\bar{m}$ its counterpart, a single fusion layer can be written as

$$z_{t,l+1}^m = \mathrm{X}_{\text{attn}}\big(\mathrm{T}_{\text{attn}}(\mathrm{S}_{\text{attn}}(z_{t,l}^m),\ h_c),\ z_{t,l}^{\bar{m}}\big) \tag{9}$$

where $\mathrm{S}_{\text{attn}}$ performs intra-modal contextualization, $\mathrm{T}_{\text{attn}}$ conditions the representation on text features, and $\mathrm{X}_{\text{attn}}$ exchanges information across modalities.

After joint modeling, the audio and video latent variables are fed into their respective prediction heads to estimate modality-specific velocity fields. The overall optimization objective is defined as

$$\mathcal{L} = \mathbb{E}_t\big[\|\hat{v}_\theta^a(z_t^a, z_t^v, h_c, t) - v_t^a\|_2^2 + \|\hat{v}_\theta^v(z_t^v, z_t^a, h_c, t) - v_t^v\|_2^2\big]. \tag{10}$$

This training strategy maintains a unified generative framework while preserving modality-specific representational

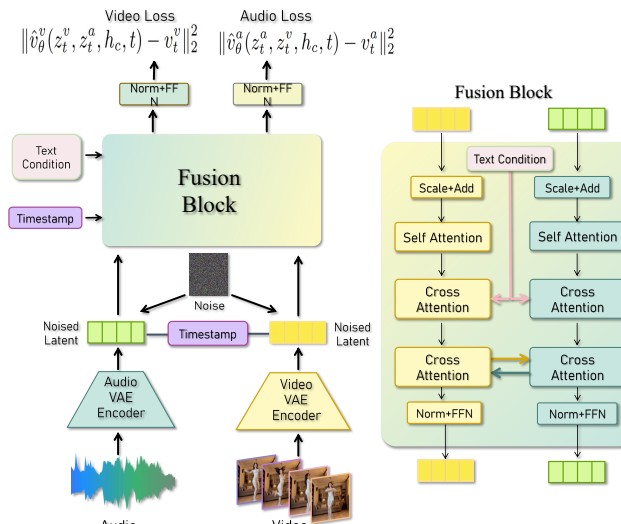

*Figure 4.* Overview of the unified diffusion architecture for text-driven music–dance generation.

spaces and prediction objectives. As a result, music and dance are jointly modeled within a single diffusion process, providing a stable foundation for cross-modal rhythmic consistency and semantic alignment.

## 5. Experiments

### 5.1. Dataset Processing

We adopt a quality-first data construction strategy and build two complementary data streams: (i) a pure-music stream for structured music semantic modeling and training the Music Captioner, and (ii) a rhythm-aligned audio–video stream for joint music–dance generation. The final dataset comprises a **10k**-scale set of curated music–dance pairs.

A detailed description of the dataset processing procedures is provided in Appendix A.

### 5.2. Experimental Settings

**Benchmarks and prompts.** We evaluate methods on the TMD-Bench test set, which consists of 100 prompts for generation, covering diverse music and dance patterns. We use **Gemini 3.0 Pro** as the MLLM evaluator throughout our evaluation pipeline, given its superior human-aligned validity and evaluation reliability (Appendix D).

**Baselines.** We group baselines into four categories: (1) closed-source end-to-end systems (Sora 2 (OpenAI, 2025), Veo 3 (DeepMind, 2025), Seedance 1.5 pro (Seed, 2025), Kling 2.6 (Kuaishou, 2025), Wan 2.6 (Group, 2025b) ), (2) cascaded text-to-dance-to-music pipelines (Wan 2.2 (Wan et al., 2025)+Hunyuan-Foley (Shan et al., 2025)), (3) cascaded text-to-music-to-dance pipelines (ACE-Step (Gong

*Table 1.* Music Captioner reliability measured by MLLM-based VQA agreement.

| Dim. | Inst. | Rhy. | Tempo | Genre | Emo. | Func. |
|---|---|---|---|---|---|---|
| Music Captioner | 0.80 | 0.79 | 0.91 | 0.82 | 0.90 | 0.93 |

et al., 2025)+Wan 2.5 (Group, 2025a)), and (4) open-source end-to-end models (JavisDiT (Liu et al., 2025), Universe-1 (Wang et al., 2025b), Ovi (Low et al., 2025), LTX-2 (HaCohen et al., 2026)). Our model is denoted as **RhyJAM**.

## 5.3. Main Results

### 5.3.1. EVALUATING THE MUSIC CAPTIONER

To assess the reliability of the Music Captioner as a source of structured semantic labels for downstream evaluation and large-scale annotation, we adopt a MLLM-based VQA-style assessment: given an audio clip and the captioner-predicted label for each semantic dimension, the judge outputs whether the label is consistent with the audio content. Table 1 reports the agreement rate across six dimensions. The captioner achieves strong agreement, especially on tempo and functional scenes with an accuracy of 0.91 and 0.93, indicating that it can serve as a stable component for fine-grained music semantic evaluation.

### 5.3.2. AUDIO EVALUATION

Evaluation of textually conditioned music generation centers on instruction adherence and generation quality. Corresponding results are summarized in Table 2.

For instruction adherence, structured semantic labels are inferred using the Music Captioner and compared against prompt-specified attributes at multiple semantic levels. As seen in Table 2, achieving faithful semantic control remains challenging for all baselines, especially for compositional attributes such as Genre, Functional Scenes, and Ambiance & Emotion, which depend on musical structure beyond local spectral cues. The proposed **RhyJAM** model demonstrates notable gains on rhythm-relevant dimensions, achieving the strongest score on Rhythm & Groove (0.59) and competitive performance on Tempo (0.51).

For generation aesthetics, closed-source commercial systems attain the highest averages, reflecting stronger production fidelity and perceptual clarity. Open-source models display larger performance variance across dimensions, suggesting less stable audio mastering and timbral consistency. Within this landscape, **RhyJAM** achieves a competitive aesthetics result, indicating stable perceptual listening quality, and strong scores in content enjoyment and usefulness. Despite this performance, a clear gap remains relative to top commercial systems, revealing considerable headroom for future advances in end-to-end music generation.

### 5.3.3. VIDEO EVALUATION

We evaluate videos using both low-level computable metrics and high-level MLLM-based judgments. As shown in Table 3, low-level video quality is summarized by three aggregated metrics: Consistency (averaging subject and background consistency), Motion (averaging dynamic degree and motion smoothness), and Quality (averaging imaging and aesthetic quality). High-level evaluation is conducted via Gemini, covering instruction following as well as holistic judgments of motion, visual quality, and consistency.

Overall, closed-source models achieve strong performance across metrics, particularly in visual quality and semantic consistency. Among open-source approaches, **RhyJAM** demonstrates a competitive and balanced profile. At the high semantic level, our model achieves competitive judgments on motion, quality, and consistency, outperforming most open-source baselines. Nevertheless, a noticeable performance gap between open-source and closed-source models remains, suggesting that further advances are still required to fully bridge this divide in dance video generation.

### 5.3.4. AUDIO–VISUAL RHYTHMIC ALIGNMENT

Audio–visual rhythmic alignment remains challenging for both open-source and closed-source generation systems. As shown in Table 4, most existing models struggle to simultaneously achieve accurate beat proximity and sufficient beat coverage. This suggests that, despite strong audio or video quality in isolation, robust cross-modal rhythmic synchronization is still inadequately addressed by current systems.

Under the algorithmic metrics, different models exhibit distinct characteristics. Closed-source models such as Sora 2 achieve high beat proximity (VBCS=0.50) and low deviation (HSD=0.12), but suffer from limited beat coverage (ABHS=0.16). Cascaded pipelines, exemplified by ACE+Wan, improve beat coverage (ABHS=0.25), yet still fall short in overall consistency. Among open-source baselines, JavisDiT shows moderate results but exhibit larger deviation values, indicating less stable temporal alignment.

In contrast, **RhyJAM** achieves the best performance by jointly preserving strong beat proximity (VBCS=0.50) and the best beat coverage among all methods (ABHS=0.27), while maintaining low temporal dispersion. The high-level perceptual evaluation further reinforces this observation: our model achieves a competitive alignment score (0.79), surpassing all open-source and cascade baselines, and reaching performance close to the strongest closed-source system. Overall, these results indicate that while both open-source and closed-source baselines exhibit limited rhythmic alignment capability, **RhyJAM** achieves consistently competitive alignment under both algorithmic and perceptual criteria.

*Table 2.* Music instruction-following and perceptual quality evaluation. We report **Low-level** and **High-level** scores for both **Sem.** (instruction semantics) and **Aes.** (aesthetic/perceptual quality).

| Method | Low-Level Metrics | | | | | High-Level Judgements | | | | | | | | | | Avg. |
|---|---|---|---|---|---|---|---|---|---|---|---|---|---|---|---|---|
| | **Sem.** | **Aes.** | | | | **Sem.** | | | | | | **Aes.** | | | | |
| | CLAP | PC | CE | PQ | CU | Inst. | Rhy. | Tempo | Genre | Amb. | Func. | MPC | MCE | MPQ | MCU | |
| *Closed-Source Model – Text-to-Music and Dance* | | | | | | | | | | | | | | | | |
| Sora 2 | 0.51 | 0.57 | 0.61 | 0.66 | 0.67 | 0.40 | 0.53 | 0.34 | 0.09 | 0.28 | 0.17 | **0.54** | **0.64** | **0.64** | **0.67** | 0.52 |
| Veo 3 | 0.53 | 0.58 | 0.75 | **0.79** | 0.80 | **0.41** | 0.57 | 0.35 | 0.13 | 0.24 | 0.14 | 0.52 | 0.58 | **0.64** | 0.62 | 0.54 |
| Seedance 1.5 pro | 0.55 | **0.70** | **0.80** | 0.71 | 0.77 | 0.26 | 0.41 | 0.37 | 0.09 | 0.25 | 0.08 | 0.44 | 0.54 | 0.63 | 0.54 | 0.52 |
| Kling 2.6 | **0.59** | 0.57 | 0.70 | 0.75 | 0.77 | 0.31 | 0.49 | 0.36 | 0.09 | 0.32 | 0.15 | 0.47 | 0.57 | 0.62 | 0.58 | 0.53 |
| Wan 2.6 | 0.52 | 0.46 | 0.55 | 0.71 | 0.69 | 0.13 | 0.47 | 0.24 | 0.07 | 0.21 | 0.10 | 0.36 | 0.42 | 0.47 | 0.43 | 0.44 |
| *Cascaded Method – Text-to-Dance-to-Music* | | | | | | | | | | | | | | | | |
| Wan 2.2+Hunyuan-Foley | 0.53 | 0.48 | 0.60 | 0.73 | 0.72 | 0.22 | 0.49 | 0.35 | 0.13 | 0.31 | 0.14 | 0.44 | 0.47 | 0.53 | 0.47 | 0.48 |
| *Cascaded Method – Text-to-Music-to-Dance* | | | | | | | | | | | | | | | | |
| ACE-Step+Wan 2.5 | 0.52 | 0.48 | 0.60 | 0.73 | 0.72 | 0.26 | 0.33 | 0.27 | 0.12 | 0.24 | 0.17 | 0.50 | 0.54 | 0.60 | 0.59 | 0.49 |
| *Open-Source Model – Text-to-Music and Dance* | | | | | | | | | | | | | | | | |
| JavisDiT | 0.51 | 0.59 | 0.62 | 0.64 | 0.69 | 0.25 | 0.34 | 0.34 | 0.08 | 0.24 | 0.18 | 0.38 | 0.40 | 0.52 | 0.48 | 0.46 |
| Universe-1 | 0.50 | 0.45 | 0.44 | 0.60 | 0.61 | 0.31 | 0.45 | 0.27 | **0.15** | 0.25 | **0.32** | 0.43 | 0.44 | 0.51 | 0.54 | 0.45 |
| Ovi | 0.54 | 0.51 | 0.76 | 0.78 | **0.82** | 0.26 | 0.48 | **0.55** | 0.09 | 0.28 | 0.17 | 0.41 | 0.51 | 0.57 | 0.52 | 0.52 |
| LTX-2 | 0.55 | 0.49 | 0.55 | 0.60 | 0.62 | 0.34 | 0.41 | 0.27 | 0.11 | **0.38** | 0.12 | 0.45 | 0.41 | 0.41 | 0.41 | 0.45 |
| **RhyJAM (Ours)** | 0.54 | 0.55 | 0.76 | 0.74 | 0.81 | 0.31 | **0.59** | 0.51 | 0.10 | 0.21 | 0.07 | 0.46 | 0.51 | 0.55 | 0.53 | 0.52 |

*Table 3.* Video generation evaluation. Low-level metrics reflect algorithmic performance, while high-level judgments evaluate perceptual consistency, motion realism, visual quality, and instruction following (IF).

| Method | Low-Level Metrics | | | | High-Level Judgments | | | | Avg. |
|---|---|---|---|---|---|---|---|---|---|
| | Cons. | Motion | Qual. | ViCLIP | Cons. | Motion | Qual. | IF | |
| *Closed-Source Model – Text-to-Music and Dance* | | | | | | | | | |
| Sora 2 (OpenAI, 2025) | 0.93 | **0.99** | 0.61 | 0.63 | 0.74 | 0.82 | 0.49 | **0.91** | 0.74 |
| Veo 3 (DeepMind, 2025) | 0.93 | 0.94 | **0.68** | 0.63 | 0.85 | 0.83 | **0.60** | 0.86 | 0.78 |
| Seedance 1.5 pro (Seed, 2025) | 0.92 | **0.99** | 0.62 | 0.62 | 0.86 | 0.86 | 0.57 | 0.71 | 0.78 |
| Kling 2.6 (Kuaishou, 2025) | **0.94** | 0.80 | 0.63 | 0.63 | 0.82 | 0.84 | 0.58 | 0.74 | 0.75 |
| Wan 2.6 (Group, 2025b) | **0.94** | 0.98 | 0.66 | **0.65** | **0.90** | **0.87** | 0.55 | 0.76 | 0.79 |
| *Cascaded Method – Text-to-Dance-to-Music* | | | | | | | | | |
| Wan 2.2 (Wan et al., 2025)+Hunyuan-Foley (Shan et al., 2025) | **0.94** | 0.95 | 0.65 | **0.65** | 0.83 | 0.82 | 0.52 | 0.72 | 0.76 |
| *Cascaded Method – Text-to-Music-to-Dance* | | | | | | | | | |
| ACE-Step (Gong et al., 2025)+Wan 2.5 (Group, 2025a) | 0.93 | **0.99** | 0.62 | 0.62 | 0.83 | **0.87** | 0.39 | 0.81 | 0.74 |
| *Open-Source Model – Text-to-Music and Dance* | | | | | | | | | |
| JavisDiT (Liu et al., 2025) | 0.90 | 0.95 | 0.47 | 0.62 | 0.36 | 0.60 | 0.03 | 0.53 | 0.53 |
| Universe-1 (Wang et al., 2025b) | 0.93 | 0.86 | 0.56 | 0.62 | 0.67 | 0.73 | 0.26 | 0.52 | 0.65 |
| Ovi (Low et al., 2025) | **0.94** | 0.92 | 0.52 | 0.63 | 0.76 | 0.75 | 0.31 | 0.58 | 0.68 |
| LTX-2 (HaCohen et al., 2026) | **0.94** | 0.90 | 0.62 | 0.62 | 0.82 | 0.83 | 0.42 | 0.52 | 0.73 |
| **RhyJAM (Ours)** | **0.94** | **0.99** | 0.58 | 0.63 | 0.73 | 0.79 | 0.37 | 0.56 | 0.71 |

### 5.3.5. CASE STUDY

To further understand how joint alignment is achieved, we examine cross-modal attention maps between the audio-video streams. Figure 5 visualizes representative attention matrices, where rows correspond to video tokens and columns correspond to audio tokens. The left panel shows our **RhyJAM** model, while the right panel depicts the baseline Ovi without explicit rhythmic alignment training.

The **RhyJAM** attention patterns exhibit smooth, continuous diagonal structures that persist across extended temporal ranges, indicating stable cross-modal correspondence between visual motion accents and musical beat progression. In contrast, the baseline attention displays fragmented activation with discontinuities and local jittering, reflecting weaker temporal coupling and reduced rhythmic coherence. These qualitative observations are consistent with the quan-

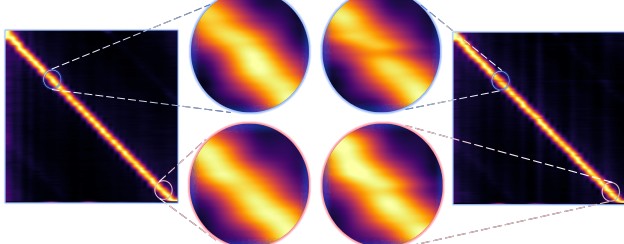

*Figure 5.* Cross-modal attention visualization. Rows denote video tokens and columns denote audio tokens. **RhyJAM** (left) exhibits smooth diagonal patterns that reflect stable audio–visual coupling.

titative results reported earlier, indicating a more coherent form of audio–visual coupling.

*Table 4.* Audio–visual rhythmic alignment evaluation.**Low-level alignment** is assessed using beat-centric algorithmic metrics. **High-level alignment** is measured by perceptual judgments of MLLM. Avg. averages the low-level score $(\text{VBCS} + \text{ABHS})/2$ with the high-level score Align.

| Method | Low-Level Alignment | | | | High-Level | Avg. |
|---|---|---|---|---|---|---|
| | VBCS↑ | CSD↓ | ABHS↑ | HSD↓ | Align.↑ | |
| *Closed-Source Model – Text-to-Music and Dance* | | | | | | |
| Sora 2 | **0.50** | 0.16 | 0.16 | **0.12** | **0.85** | 0.59 |
| Veo 3 | 0.45 | 0.17 | 0.22 | 0.17 | 0.84 | 0.59 |
| Seedance 1.5 pro | 0.47 | 0.21 | 0.19 | 0.15 | 0.77 | 0.55 |
| Kling 2.6 | 0.43 | 0.16 | 0.25 | 0.11 | 0.73 | 0.54 |
| Wan 2.6 | 0.40 | 0.28 | 0.17 | 0.16 | 0.59 | 0.44 |
| *Cascaded Method – Text-to-Dance-to-Music* | | | | | | |
| Wan+Foley | **0.50** | **0.14** | 0.25 | 0.13 | 0.77 | 0.57 |
| *Cascaded Method – Text-to-Music-to-Dance* | | | | | | |
| Ace+Wan | 0.41 | 0.18 | 0.25 | 0.13 | 0.68 | 0.51 |
| *Open-Source Model – Text-to-Music and Dance* | | | | | | |
| JavisDiT | 0.46 | 0.22 | 0.23 | 0.19 | 0.66 | 0.50 |
| Universe-1 | 0.42 | 0.19 | 0.20 | 0.13 | 0.63 | 0.47 |
| Ovi | 0.30 | 0.22 | 0.22 | 0.19 | 0.69 | 0.48 |
| LTX-2 | 0.34 | 0.20 | 0.21 | 0.14 | 0.71 | 0.49 |
| **RhyJAM (Ours)** | **0.50** | 0.19 | **0.27** | **0.12** | 0.79 | 0.59 |

# 6. Conclusion

We present **TMD-Bench**, a benchmark for text-driven music–dance co-generation that evaluates unimodal quality, instruction semantics, and rhythmic alignment using physical metrics and MLLM-based judgments. We hope TMD-Bench will spur progress toward stronger rhythmic and kinetic coherence in next-generation generative systems.

# Acknowledgements

We thank the anonymous reviewers for their constructive comments and valuable suggestions. We also thank our colleagues and collaborators for helpful discussions and feedback. This work was supported by National Natural Science Foundation of China under Grant No.U25B2064 and National Natural Science Foundation of China under Grant No.U24A20326.

# Impact Statement

This paper presents work whose goal is to advance the field of Machine Learning. There are many potential societal consequences of our work, none which we feel must be specifically highlighted here.

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

# A. Dataset Construction Details

Our benchmark is constructed under a quality-first philosophy and consists of two complementary data streams: (i) a pure-music stream for structured music semantic modeling and captioning, and (ii) an audio–video stream for rhythmic alignment and joint music–dance generation.

**Pure-music data processing.** We construct a high-quality music corpus through an iterative refinement pipeline. Specifically, we (1) annotate audio clips using an MLLM-assisted labeling pipeline, (2) perform human verification and correction, and (3) train a Music Captioner on the cleaned labels. The trained captioner is then used to re-annotate larger-scale unlabeled audio, followed by an additional round of human correction. We repeat this *caption–verify–retrain* loop to progressively improve both label quality and captioner reliability, yielding a robust semantic backbone for music instruction-following evaluation and large-scale audio annotation.

**Audio–video data processing for rhythmic alignment.** For the joint audio–video stream, we curate dance-centric videos and enforce strict audio quality and rhythm constraints. Concretely, we first filter candidate clips to retain dance videos with clear human motion and stable framing. We then apply UVR5 to separate vocals from accompaniment in order to reduce vocal interference for beat-centric rhythm analysis. Next, we generate both music captions and video captions for semantic screening. To favor rhythm-sensitive music, we filter clips by music-caption fields (e.g., rhythm/groove descriptors) and apply RMS-based silence removal to exclude near-silent segments. We further use SNR-based filtering to retain high-quality accompaniment signals. Finally, all retained samples undergo human review to ensure (i) clear beat structure, (ii) salient motion accents, and (iii) reliable audio–video synchronization. This procedure yields a 10k-scale rhythm-aligned music–dance dataset that supports stable multimodal learning and evaluation.

**Data sources and coverage.** Our 10k-scale music–dance pairs are collected from publicly available online platforms (e.g., YouTube and similar video-sharing websites), covering diverse dance styles, performer settings, and music genres. During curation, we aim to maintain broad category coverage such that both training and test splits contain diverse and representative combinations of music attributes, dance styles, and scene contexts.

# B. Model Implementation Details

We train our model RhyJAM using DeepSpeed ZeRO-2 with bf16 mixed precision. We use AdamW with a learning rate of 1e-5, weight decay 0.01, and a constant learning-rate schedule. We apply 8 gradient-accumulation steps. Training is run for 10 epochs and checkpoints. For preprocessing, audio is resampled to 16 kHz. Videos are center-cropped and resized to 480×480, sampled at 24 fps, with 117 frames per clip. We train with 1000 training timesteps (Flow-Matching scheduler, shift 5). For inference, we run 50 sampling steps (UniPC solver, shift 5) and apply classifier-free guidance with separate scales for audio and video (3.0 and 4.0, respectively).

# C. Empirical Analysis of Beat-Centric Rhythmic Metrics

MDAlign builds on two beat-centric metrics, VBCS and ABHS, alongside their empirical standard deviations CSD and HSD. To examine whether these metrics capture perceptual rhythmic alignment and reflect properties of the underlying data, we compute them on both model outputs and subsets of the training dataset.

**Model outputs vs. training dataset.** We first compare RhyJAM outputs on the evaluation set with a random 1k subset of the 10k-scale rhythm-aligned training dataset (Table 5). The dataset subset achieves higher VBCS/ABHS and lower dispersion than the model outputs, indicating that the curated data exhibit a stronger rhythmic prior than what the current model fully reproduces. This suggests that the training data distribution encodes meaningful beat–motion coupling.

**High-alignment subset and metric sensitivity.** To verify that the metrics track perceptual rhythmic quality, we additionally select 30 clips from the dataset that humans consistently judge as strongly on-beat. Relative to both model outputs and the random subset, this curated subset shows monotonically higher VBCS/ABHS, with dispersion remaining at a similar scale. This monotonic separation aligns with human preference ordering and suggests that VBCS/ABHS provide a workable scalarization of the rhythm-alignment objective.

*Table 5.* Beat-centric rhythmic statistics on model outputs and dataset subsets. Higher VBCS/ABHS indicate better beat alignment; lower CSD/HSD indicate lower dispersion.

| Split | VBCS↑ | ABHS↑ | CSD↓ | HSD↓ |
|---|---|---|---|---|
| **RhyJAM** | 0.50 | 0.27 | 0.19 | 0.12 |
| Random 1k from 10k dataset | 0.55 | 0.35 | 0.14 | 0.13 |
| 30 curated high-alignment clips | 0.62 | 0.41 | 0.17 | 0.15 |

# D. Selection and Validation of MLLM-as-a-Judge

To ensure the reliability of the high-level perceptual evaluation within the TMD-Bench framework, we conducted a rigorous comparative study to select the most suitable MLLM as our automated judge. We evaluated four representative models: Gemini 3.0 Pro, Gemini 3.0 Flash , Gemini 2.5 Pro, and Qwen 3 Omni.

## D.1. Alignment with Human Judgment

We invited ten undergraduate students to independently rate 100 generated music–dance videos across the 12 evaluation dimensions defined in Section 3. Raters were intentionally diverse, including five students with a computer-science background and five from non-CS majors. The 100 videos were selected to cover a wide range of dance styles, music genres, and scene contexts, and were sampled in a balanced manner with a similar number of clips from each baseline to avoid skew. To reduce potential information leakage that could make the source model identifiable, we standardized all evaluation videos to the same resolution and duration, and removed model-specific visual cues such as watermarks or other distinctive overlays whenever present.

We then calculated the correlation between the MLLM-generated scores and the mean human scores using three metrics: Pearson Linear Correlation Coefficient (PLCC), Spearman Rank Correlation Coefficient (SRCC), and Quadratic Weighted Kappa (QWK).

Table 6 presents the dimension-wise correlation results, while Table 7 provides the macro-averaged performance for each model. The results indicate that Gemini 3.0 Pro significantly outperforms other models in the majority of categories. While certain models may exhibit competitive performance in isolated dimensions, Gemini 3.0 Pro demonstrates the most robust and balanced capacity to evaluate complex multimodal content, achieving the highest overall agreement with human judgment across visual consistency and auditory aesthetic metrics.

*Table 6.* Detailed correlation between MLLM scores and human judgments across 12 dimensions.

| Metric | Gemini 3.0 Pro | | | Gemini 2.5 Pro | | | Gemini 3.0 Flash | | | Qwen 3 Omni | | |
|---|---|---|---|---|---|---|---|---|---|---|---|---|
| | PLCC | SRCC | QWK | PLCC | SRCC | QWK | PLCC | SRCC | QWK | PLCC | SRCC | QWK |
| AQ | 0.386 | 0.353 | 0.230 | 0.254 | 0.117 | 0.091 | 0.205 | 0.203 | 0.077 | 0.228 | 0.236 | 0.032 |
| BC | 0.685 | 0.656 | 0.384 | 0.005 | 0.048 | -0.034 | 0.306 | 0.234 | 0.241 | 0.005 | 0.048 | -0.034 |
| CE | 0.776 | 0.776 | 0.318 | 0.338 | 0.324 | 0.036 | 0.178 | 0.186 | 0.024 | 0.040 | 0.005 | 0.042 |
| CU | 0.745 | 0.756 | 0.367 | 0.305 | 0.297 | 0.091 | 0.005 | 0.039 | -0.012 | -0.037 | -0.063 | -0.048 |
| DD | 0.529 | 0.484 | 0.170 | 0.528 | 0.489 | 0.262 | 0.027 | 0.011 | -0.027 | 0.274 | 0.226 | 0.014 |
| IF | 0.445 | 0.395 | 0.233 | 0.308 | 0.320 | 0.185 | 0.460 | 0.401 | 0.229 | 0.357 | 0.393 | 0.168 |
| IQ | 0.637 | 0.604 | 0.189 | 0.310 | 0.222 | 0.093 | -0.083 | -0.051 | -0.051 | 0.347 | 0.256 | 0.034 |
| MS | 0.632 | 0.688 | 0.384 | 0.089 | 0.098 | 0.074 | 0.138 | 0.231 | 0.140 | -0.153 | -0.167 | -0.040 |
| PC | 0.620 | 0.629 | 0.161 | 0.202 | 0.229 | 0.028 | 0.096 | 0.121 | 0.009 | 0.088 | 0.086 | 0.116 |
| PQ | 0.743 | 0.741 | 0.310 | 0.035 | 0.109 | 0.031 | -0.108 | -0.055 | -0.110 | -0.104 | -0.082 | -0.054 |
| SC | 0.593 | 0.596 | 0.537 | 0.374 | 0.315 | 0.161 | 0.118 | 0.113 | 0.027 | 0.228 | 0.206 | 0.010 |
| VA | 0.549 | 0.422 | 0.372 | 0.558 | 0.514 | 0.410 | 0.288 | 0.241 | 0.100 | 0.023 | 0.021 | 0.049 |

## D.2. Stability and Self-Consistency Analysis

A critical requirement for an automated judge is reproducibility. We analyzed the stability of Gemini 3.0 Pro and Gemini 2.5 Pro by performing 50 independent scoring runs on the same 100 test videos. Stability was quantified using normalized

*Table 7.* Overall average correlation metrics across all evaluated models.

| Model | PLCC | QWK | SRCC |
|---|---|---|---|
| Gemini 3.0 Flash | 0.1358 | 0.0540 | 0.1394 |
| Gemini 3.0 Pro | **0.6117** | **0.3045** | **0.5916** |
| Gemini 2.5 Pro | 0.2756 | 0.1190 | 0.2568 |
| Qwen-Omni | 0.1081 | 0.0240 | 0.0970 |

entropy-based consistency:

$$C = 1 - \frac{-\sum_{k=1}^{5} p_k \log p_k}{\log 5} \tag{11}$$

where $p_k$ represents the empirical distribution of the scores across the 5 possible rating levels $\{1, 2, 3, 4, 5\}$.

As shown in Table 8, Gemini 3.0 Pro exhibits exceptional stability, achieving perfect consistency ($C = 1.0$) in 11 out of 12 dimensions. In contrast, Gemini 2.5 Pro shows lower reliability, particularly in subjective auditory metrics such as Production Complexity ($C = 0.832$) and Content Enjoyment ($C = 0.874$). The near-deterministic nature of Gemini 3.0 Pro's reasoning process ensures that the TMD-Bench results remain consistent across repeated evaluations.

*Table 8.* Comparison of stability between Gemini 2.5 Pro and Gemini 3.0 Pro across 50 repeated evaluations.

| Metric Dimension | Gemini 2.5 Pro | Gemini 3.0 Pro |
|---|---|---|
| Instruction Following (IF) | 1.0000 | **1.0000** |
| Rhythmic Alignment (VA) | 0.9798 | **1.0000** |
| Subject Consistency (SC) | 1.0000 | **1.0000** |
| Dynamic Degree (DD) | 1.0000 | **1.0000** |
| Background Consistency (BC) | 1.0000 | **1.0000** |
| Motion Smoothness (MS) | 1.0000 | **1.0000** |
| Imaging Quality (IQ) | 0.9689 | **1.0000** |
| Audio Quality (AQ) | 0.9378 | **0.9798** |
| Production Complexity (PC) | 0.8327 | **1.0000** |
| Content Enjoyment (CE) | 0.8745 | **1.0000** |
| Production Quality (PQ) | 0.8853 | **1.0000** |
| Content Usefulness (CU) | 0.8508 | **1.0000** |

Based on its superior correlation with human perception and its robust self-consistency, we select Gemini 3.0 Pro as the core evaluation engine for the TMD-Bench framework.

# E. MLLM Prompt Template

For reproducibility of our MLLM-as-a-Judge evaluation, we provide the full prompt templates used to query the Gemini-based evaluators in Figures 6 to 10. These templates explicitly define the role, input format, reasoning steps, and output schema for each assessment track, ensuring that the model follows a consistent protocol across all test samples.

Specifically, the evaluation relies on the following five prompt templates:

- **Visual Audio Alignment**: this prompt (see Figure 6) is used in the MDAlign perceptual track to assess rhythmic synchronization between motion accents and musical beats.

- **Video Instruction Following**: the corresponding template (see Figure 7) measures how faithfully the generated video executes the textual instruction in terms of subjects, actions, and scene semantics.

- **Video Visual Quality**: the prompt (see Figure 8) evaluates both imaging quality and aesthetic quality as a two-dimensional perceptual score.

- **Video Motion**: this prompt (see Figure 9) focuses on subject consistency, background consistency, motion smoothness, and dynamic degree.

- **Auditory Aesthetic**: the audio evaluation template (see Figure 10) assesses production complexity, content enjoyment, production quality, and content usefulness using four MOS-style metrics.

Together, these templates operationalize the multi-track evaluation framework described in the main text, and can be directly reused to replicate our MLLM-based judgments or to extend TMD-Bench to new models in a fully standardized way.

## F. Case Visualization

To complement the quantitative results of Our model (RhyJAM) reported in the main paper, we provide qualitative examples of text-driven music–dance co-generation in Figures 11 to 13. Each case is selected from the TMD-Bench evaluation set and visualizes representative prompt–output pairs that cover diverse subjects, dance styles, scene contexts, and musical characteristics. For each example, we show sampled video frames, the corresponding audio waveform, and high-level semantic tags, allowing visual inspection both unimodal fidelity and cross-modal rhythmic coupling.

## Visual Audio Alignment

You are a strict video-audio alignment evaluator for music/dance videos.You will Evaluate how well the VISUAL motion rhythm aligns with the AUDIO rhythm.

You will be given:
1. **Video B**: the generated video
--------------------------------------------------------------------
## Key Definitions
- Audio beats: salient rhythmic pulses in the soundtrack (what a listener would clap/tap to).
- Motion beats: salient movement accents (peaks of motion intensity, sharp direction changes, footfalls, body hits, or clear pose transitions).
- Alignment: perceptual temporal coherence between audio beats and motion beats.
--------------------------------------------------------------------
## Reasoning Steps
### 1. Alignment Assessment
Evaluate temporal relationship between audio beats and motion accents:
- Timing proximity: do motion hits land near audio beats?
- Coverage: how many audio beats are answered by motion accents?
- Stability: is alignment consistent, or only present briefly?
- Errors: systematic early/late offsets, drift, or random mismatches?
--------------------------------------------------------------------
### 2. Decision
- **5 – Excellent**: Extremely tight synchronization; motion accents reliably land on or very near beats; drift and timing errors are minimal; sustained alignment throughout the clip.
- **4 – Strong**: Good synchronization but not perfect; mostly aligned with occasional jitter, misses, or partial drift; still clearly beat-aware.
- **3 – Mixed/Moderate**: Partial alignment; some segments synced and others off; intermittent or unstable alignment; common in generic dance videos.
- **2 – Weak**: Sparse or unreliable alignment; motion rarely responds to audible beats; may show only vague rhythmic awareness.
- **1 – Poor/None**: Mostly unaligned; motion does not correspond to beat structure; OR video is silent/no audio; OR no meaningful rhythmic correspondence is detected.
--------------------------------------------------------------------
## Output Format
Return ONLY one JSON object with the following fields:
{
  "va_alignment": A,
  "reasoning": "..."
}

*Figure 6.* Visual Audio Alignment prompt used for MLLM-based evaluation. The template defines key concepts, step-by-step reasoning, and a 1–5 alignment score for rhythmic synchronization.

**Video Instuction Following**

You are a professional digital artist and video instruction-following evaluation specialist.You will evaluate whether the video faithfully follows a given instruction.This task focuses strictly on instruction compliance.

You will be given:
1. **Video B**: the generated video.
2. **Generation Instruction**: a detailed textual description specifying subjects, actions, environment, interactions, and constraints.
--------------------------------------------------------------------
## Reasoning Steps
You must follow all steps below before assigning a score.
### 1. Detect Difference (Pure Observation)
Describe what is observed in the generated video across time:
- Number of people present     - Apparent age group, gender, and ethnicity
- Environment and setting        - What actions occur and how they evolve over time
- For multi-person videos: whether subjects interact, synchronize, mirror, or move independently

This step must NOT reference the instruction.
--------------------------------------------------------------------
### 2. Expected Video Caption (Ideal Outcome)
Write a factual description of how the video should look if the instruction were perfectly followed:
- Subject count and attributes        - Environment and spatial context
- Required dance style and characteristic movements     - Interaction pattern (if multiple dancers)
--------------------------------------------------------------------
### 3. Instruction Match Analysis
Compare Step 1 with Step 2 and evaluate:
- Are the correct subjects present (count, age group, ethnicity, gender)?
- Is the specified environment correctly depicted?
- Does the observed motion semantically match the named dance style?
- Are the required actions sustained throughout most of the video?
- For multi-person videos, Is there genuine coordination, mirroring, or formation logic?
--------------------------------------------------------------------
### 4. Decision
Assign an instruction-following score from 1 to 5 based on strict compliance.
- **5 – Perfect Compliance**: All specified subjects, actions, interactions, environments, and constraints are present and accurate throughout the video.
- **4 – Minor Omission**: The core instruction is fulfilled, but a small temporal, stylistic, or interaction detail is slightly inaccurate or incomplete.
- **3 – Partial Compliance**: The main idea is present, but one or more key aspects are incorrect, missing, weakly expressed, or inconsistently sustained.
- **2 – Major Omission**: Most required elements are missing, incorrect, or briefly appear without persistence.
- **1 – Non-Compliance**: The instruction is ignored, misinterpreted, or replaced by unrelated content.
--------------------------------------------------------------------
## Output Format
Provide the final evaluation in the following JSON format only:
{
  "instruction_score": X,
  "reasoning": "1. Detect Difference ... 2. Expected Video Caption ... 3. Instruction Match ... 4. Decision ..."
}

*Figure 7.* Video Instruction Following prompt. The judge first describes the observed video, then constructs an ideal caption from the text instruction, and finally rates semantic compliance on a 1–5 scale with accompanying reasoning.

## Video Visual Quality

You are a professional video quality assessment specialist.You will evaluate the perceptual the video.

You will be given:
1. **Video B**: the generated video.
------------------------------------------------------------------
You will score two dimensions:
1. imaging quality: technical and perceptual visual fidelity
2. aesthetic quality: artistic appeal and overall aesthetics
------------------------------------------------------------------
## Reasoning Steps
### 1. Observation (no scoring yet)
Briefly describe:
- sharpness / detail / aliasing        - noise / compression artifacts / banding
- stability (flicker, exposure/white-balance shifts)     - lighting and color consistency
- any visible model failures (warping, melting, inconsistent geometry)
------------------------------------------------------------------
### 2.  Dimension Analysis
- imaging_quality: assess clarity, detail, artifact level, temporal stability of rendering, and overall technical fidelity.
- aesthetic_quality:assess composition, color harmony, lighting aesthetics, style coherence, and "pleasantness" as a final video.
------------------------------------------------------------------
### 3.  Decision
Give two integer scores from 1 to 5 for the two dimensions.
**imaging quality**:
- **5**: crisp details, minimal artifacts, stable rendering across frames
- **4**: mostly clean, minor artifacts or mild softness/flicker
- **3**: noticeable artifacts/softness/compression or intermittent flicker that affects viewing
- **2**: heavy artifacts (warping, strong noise, severe blur, frequent flicker) but still recognizable
- **1**: severely broken visuals, major corruption, or persistent failures that dominate the video

**aesthetic quality**:
- **5**: strong composition/lighting/color, coherent style, visually pleasing
- **4**: good-looking overall, minor aesthetic issues
- **3**: average aesthetics; workable but not engaging, or inconsistent style
- **2**: unpleasant or messy composition/color/lighting; aesthetics clearly weak
- **1**: visually off-putting; chaotic or incoherent aesthetics throughout
------------------------------------------------------------------
## Output Format
Return ONLY this JSON object:
{
  "imaging_quality": X,
  "aesthetic_quality": Y,
  "reasoning": "..."
}

*Figure 8.* Video Visual Quality evaluation prompt. The template guides the model to score both imaging quality and aesthetic quality, capturing technical fidelity and overall visual appeal.

## Video Motion

You are a professional video motion and temporal-consistency assessment specialist.You will evaluate SPATIOTEMPORAL CONSISTENCY & MOTION QUALITY of the video.

You will be given:
1. **Video B**: the generated video.
---------------------------------------------------------------------
You will score four dimensions:
1. subject consistency: identity/appearance consistency of the main subject(s) across time
2. background consistency: scene/layout consistency and absence of temporal "teleporting"
3. motion smoothness: smoothness and physical continuity of motion (no jitter, stutter, sudden pops)
4. dynamic degree: richness and strength of motion dynamics (not static, not frozen; meaningful movement)
---------------------------------------------------------------------
## Reasoning Steps
### 1. Observation (no scoring yet)
Describe across time:
- Does the main subject keep consistent identity (face/body/clothes/shape)?
- Does the background stay stable (layout, objects, perspective)?
- Are there jumps, jitters, warps, melting, popping frames?
- Is the video mostly static or does it show continuous motion?
---------------------------------------------------------------------
### 2. Dimension Analysis
- subject_consistency: check identity stability, clothing/shape persistence, absence of sudden morphing.
- background_consistency: check camera/world stability, scene geometry, object permanence.
- motion_smoothness: check temporal continuity, frame-to-frame smoothness, absence of stutter/pops/jitter.
- dynamic_degree: assess whether motion is sufficiently present, varied, and sustained (not frozen, not minimal).
---------------------------------------------------------------------
### 3. Decision
Give four integer scores from 1 to 5 for the four dimensions.
**subject consistency**:
5: subjects remain stable; only minor natural variation    4: mostly stable; rare mild warping or brief inconsistency
3: noticeable identity/shape drift or intermittent warping   2: frequent morphing/instability; identity not reliable
1: severe instability; subject identity breaks continuously

**background consistency**:
5: stable scene/layout; coherent camera/world over time    4: mostly stable; minor flicker or small geometry drift
3: background shifts, object popping appears intermittently 2: frequent scene warps or object permanence  breaks
1: chaotic background; scene changes dominate the video

**motion smoothness**:
5: smooth and continuous motion; no obvious temporal artifacts    4: minor jitter or occasional small pops
3: noticeable stutter/jitter/popping affects perception   2: frequent temporal discontinuities; motion looks broken
1: motion is severely corrupted or largely unusable

**dynamic degree**:
5: sustained, varied motion with clear dynamics   4: good motion presence; slightly repetitive or moderate intensity
3: some motion exists but weak, or partially frozen    2: very low motion; mostly static with small movements
1: essentially frozen or near-static throughout
---------------------------------------------------------------------
## Output Format
Return ONLY this JSON object:
{
  "subject_consistency": A,
  "dynamic_degree": B,
  "background_consistency": C,
  "motion_smoothness": D,
  "reasoning": "..."
}

19

*Figure 9.* Prompt used to rate **video motion and temporal consistency**.

## Auditory Aesthetic

You are an expert audio evaluation specialist. You will rate an audio clip with FOUR MOS-style metrics.

You will be given:
1. **Audio A**: the generated audio clip.
-------------------------------------------------------------------
You will rate the audio with **FOUR MOS-style metrics**:
1. PC (Production Complexity): The structural and layering complexity of the audio.
2. CE (Content Enjoyment): The subjective pleasantness and engagement of the listening experience.
3. PQ (Production Quality): The technical cleanliness, mixing quality, and overall audio fidelity.
4. CU (Content Usefulness): The practical usability of the audio as a reusable content asset.
-------------------------------------------------------------------
## Reasoning Steps
### 1. Observation (no scoring yet)
Describe what you hear across time:
- Layering / instrumentation / structural complexity
- Enjoyment / listenability
- Noise / clipping / artifacts / distortion
- Whether it is usable as music/SFX/ambient content
-------------------------------------------------------------------
### 2. Decision
**Production Complexity**
  - **5 – Highly Elaborate**: Multiple concurrent elements, sophisticated transitions, rich arrangement/mixing.
  - **4 – Rich & Structured**: Clear sections/transitions with layered instrumentation and evolving design.
  - **3 – Moderately Layered**: Noticeable arrangement or layering with some variation over time.
  - **2 – Simple & Repetitive**: Basic looping with few layers and minimal structural variation.
  - **1 – Extremely Simple**: Near-monotone or single-layer content with almost no structural complexity.
**Content Enjoyment**
  - **5 – Highly Enjoyable**: Engaging, memorable, and professionally pleasing to listen to.
  - **4 – Enjoyable**: Pleasant and engaging with minimal distracting aspects.
  - **3 – Average Enjoyment**: Acceptable enjoyment with neutral or mildly pleasant qualities.
  - **2 – Weak Enjoyment**: Hard to enjoy, noticeably dull or problematic.
  - **1– Unpleasant**: Irritating, chaotic, or actively unpleasant to listen to.
**Production Quality**
  - **5 – Studio-Grade Quality**: Extremely clean, balanced, and well-mixed with no significant issues.
  - **4 – Clean & Balanced**: Good fidelity with only minor technical artifacts.
  - **3 – Acceptable Quality**: Usable audio with noticeable but tolerable artifacts.
  - **2 – Low Quality**: Clear technical defects, noise, or strong artifacts.
  - **1 – Poor Quality**: Severe distortion, clipping, or degradation.
**Content Usefulness**
  - **5 – Excellent Usefulness**: Immediately usable as music/SFX/ambient content across contexts.
  - **4 – Good Usefulness**: Suitable for practical usage with no major adjustments.
  - **3 – Usable with Caveats**: Can be used, but may require editing or context alignment.
  - **2 – Weak Usefulness**: Requires significant fixes or contextual support to be usable.
  - **1 – Unusable**: Fails to function as a standalone audio asset.
-------------------------------------------------------------------
## Output Format
Return **ONLY** this JSON object:
{
 "PC": A,   "CE": B,    "PQ": C,   "CU": D, "reasoning": "..."
}

*Figure 10.* Prompt for **auditory aesthetics**. The judge outputs four MOS-style scores—production complexity, content enjoyment, production quality, and content usefulness.

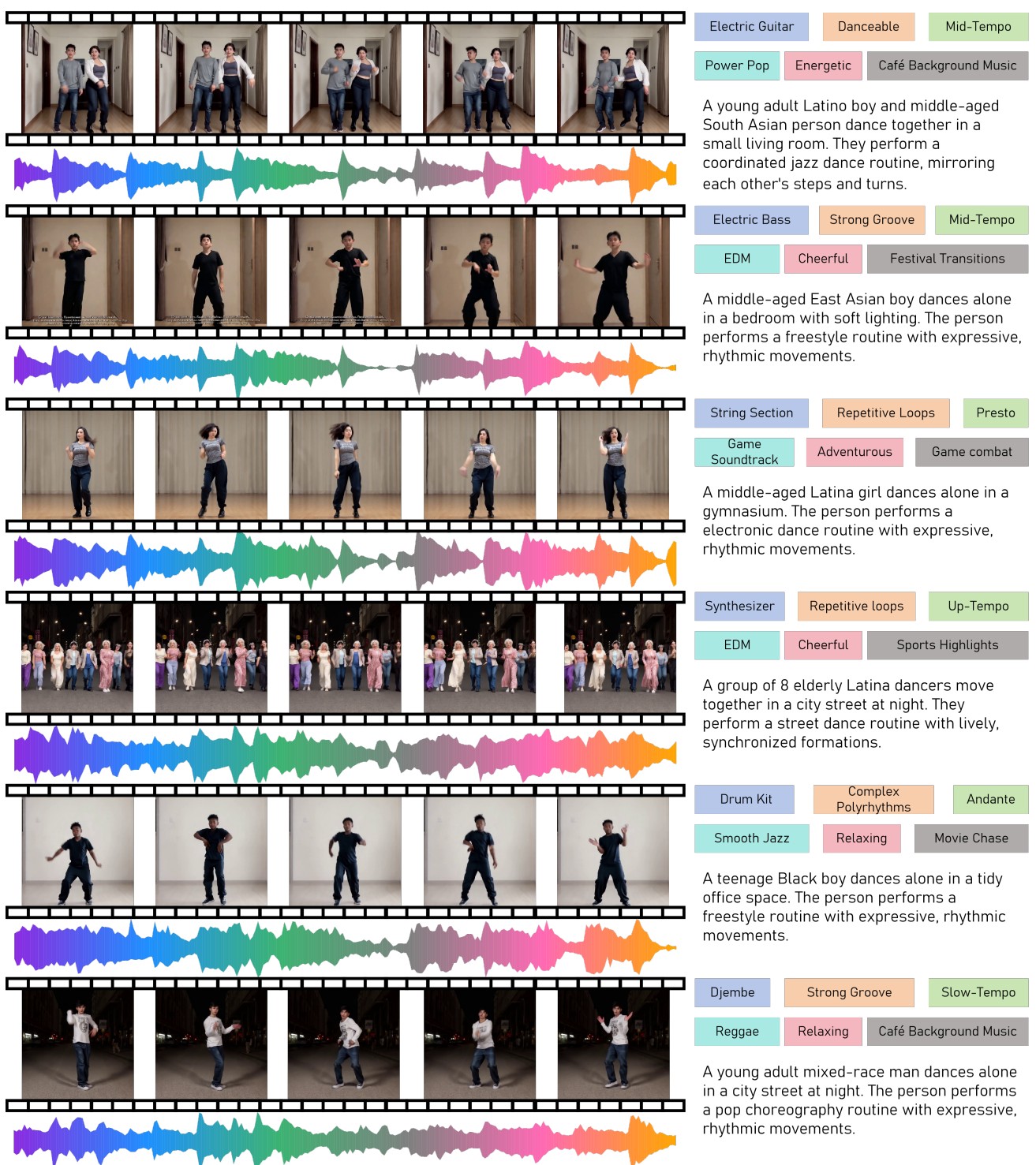

*Figure 11.* Qualitative examples of text-driven music–dance co-generation (Case 1). Each row depicts sampled video frames, the corresponding audio waveform, and semantic tags, illustrating identity consistency and rhythm-aware motion under varying prompts.

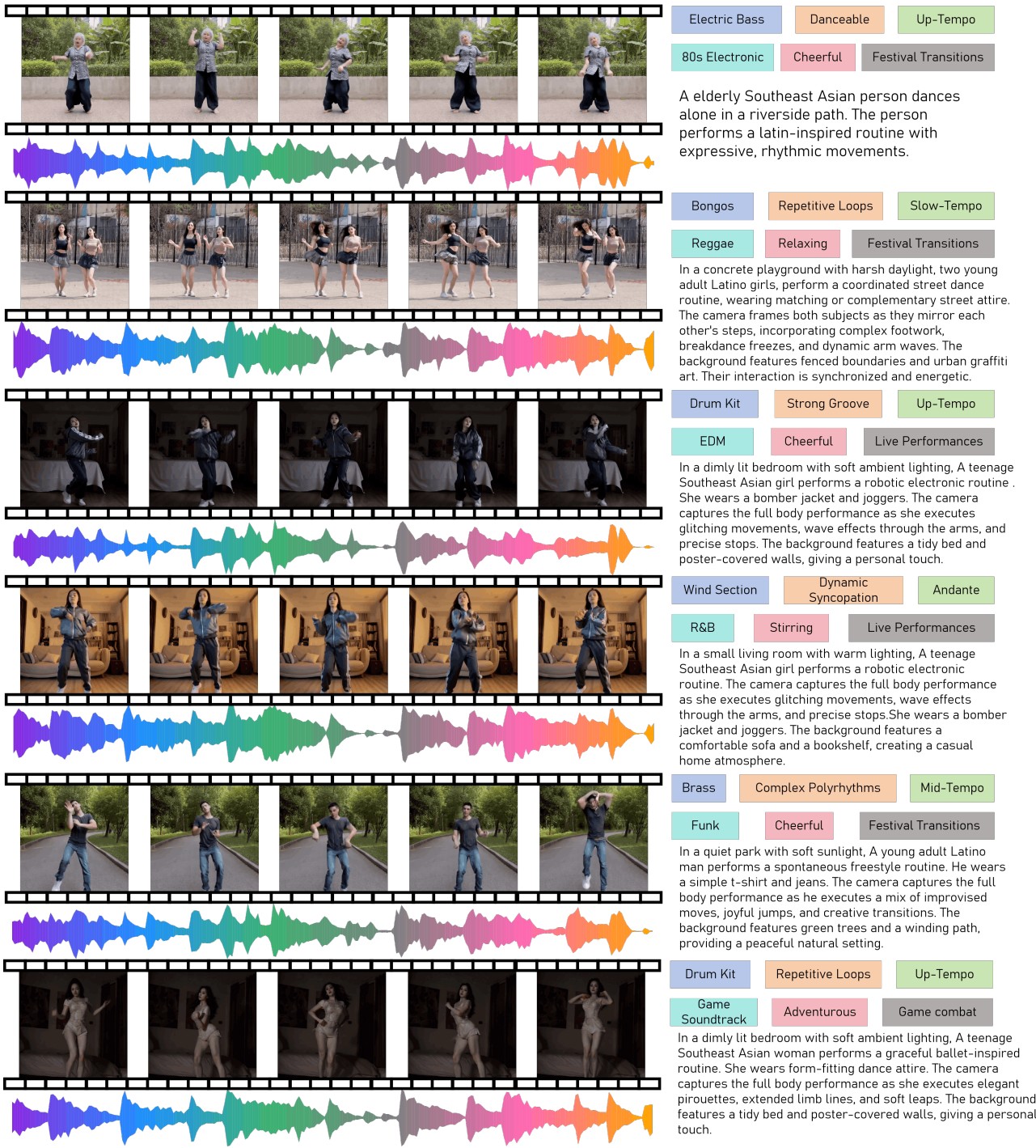

*Figure 12.* Qualitative examples of text-driven music–dance co-generation (Case 2). Each row depicts sampled video frames, the corresponding audio waveform, and semantic tags, illustrating identity consistency and rhythm-aware motion under varying prompts.

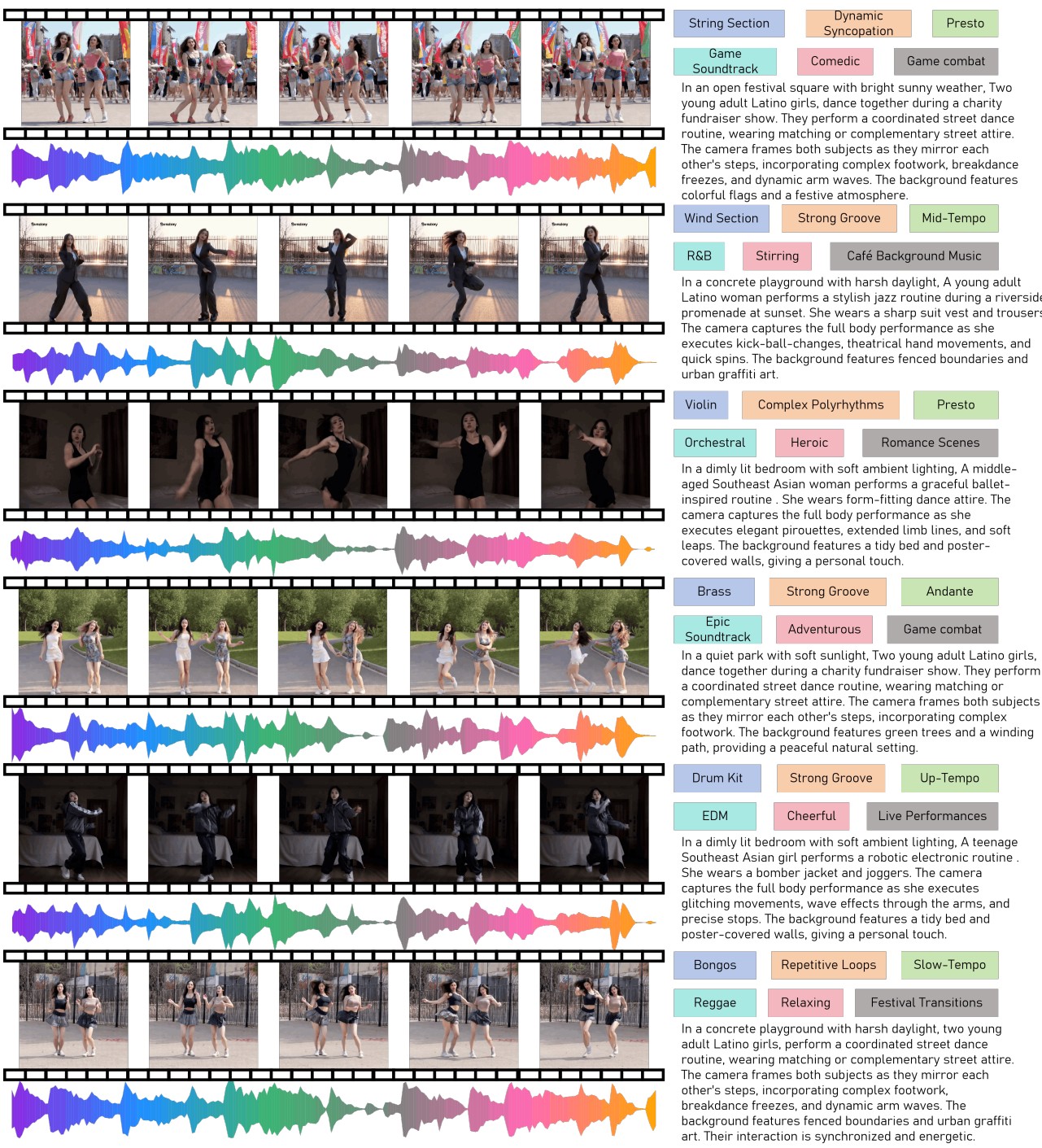

*Figure 13.* Qualitative examples of text-driven music–dance co-generation (Case 3). Each row depicts sampled video frames, the corresponding audio waveform, and semantic tags, illustrating identity consistency and rhythm-aware motion under varying prompts.

