# OpenReview forum: "TMD-Bench: A Multi-Level Evaluation Paradigm for Music–Dance Co-Generation"
_ICML.cc/2026/Conference — ICML 2026 regular_

### Official Review · Reviewer_qjM4 · 2026-02-20

**Soundness:** 2
**Presentation:** 3
**Significance:** 4
**Originality:** 3
**Overall Recommendation:** 4
**Confidence:** 4

**Summary:**

This paper presents TMD-Bench, the first multi-level benchmark for evaluating text-driven music-dance co-generation, addressing the lack of standardized assessments. It evaluates systems on unimodal quality (audio/video fidelity), instruction adherence (text-content matching), and cross-modal rhythm alignment (motion-music synchronization). The innovative MDAlign framework decomposes rhythm into low-level physical beat-motion matching (via VBCS/ABHS metrics) and high-level perceptual coherence using multimodal large models (MLLMs). To foster community progress, the authors open-source a 10,000-sample aligned dataset, a fine-grained music captioner, and a robust end-to-end baseline model, RhyJAM.

**Compliance With Llm Reviewing Policy:**

Affirmed.

**Final Justification:**

Thank you for the authors’ response. I will maintain my original score.

**Key Questions For Authors:**

See weaknesses

**Limitations:**

Yes

**Strengths And Weaknesses:**

### Strengths:
- Proposed the first comprehensive benchmark TMD-Bench for text-driven music-dance joint generation, effectively filling the evaluation gap in this field.
- Accompanied by open-sourcing a 10,000-scale aligned dataset, fine-tuned music annotator, and baseline model RhyJAM, which has extremely high practical value for the research community.
- The innovative MDAlign dual-track framework cleverly transforms the abstract "sense of rhythm" into computable physical event matching and large model perceptual adjudication, with a very reasonable design.

### Weaknesses:
- The paper completely lacks playable multimedia samples (such as anonymous web links), making it impossible for reviewers to qualitatively assess the fluency of generated actions, rhythm dynamics, and to intuitively measure the actual quality and annotation effects of the dataset.
- The data distribution diagram in the upper right corner of Figure 1 has too low resolution, blurry when enlarged.
- The physical alignment metric relies solely on 2D keypoint velocity, completely unable to capture 3D depth motion, dance freezes like "static tension," and the beat hierarchy of music.
- The cascaded baseline used for comparison relies on general video models, lacking direct comparison with professional "music-to-dance" generation models, weakening the persuasiveness of the end-to-end model's advantages.

---

> ### Author Rebuttal · Authors · 2026-03-31
>
> ## **Q1: About Demo Page**
> We thank the reviewer for this suggestion. We agree that playable multimedia samples are important for intuitive evaluation.We have provided an anonymous demo page with representative generated results and dataset examples, which can be accessed at: https://anonymous-2312.github.io/TMD-Bench/
>
> ## **Q2: About Figure Clarity**
> We thank the reviewer for pointing this out. We acknowledge that the data distribution diagram in the top-right corner of Figure 1 suffers from over-compression, which leads to reduced clarity when enlarged. In the revised version, we will replace it with a higher-resolution image and adjust the figure export settings to avoid excessive compression, ensuring improved readability.
>
> ## **Q3: About Metric**
> Our metric adopts 2D keypoint velocity as a simple and robust proxy, while avoiding the instability of monocular 3D estimation in in-the-wild settings. Although velocity-based signals may not fully capture expressive pauses (e.g., static tension), our formulation measures temporal consistency over motion dynamics rather than instantaneous magnitude, partially reflecting such effects. We will clarify these limitations in the revision and note that our framework can be naturally extended to incorporate 3D pose and multi-level beat annotations in future work.
>
> Additionally, we introduce downbeat-level metrics to complement beat-level evaluation and enrich the hierarchical modeling of rhythmic structure. This extension provides a more comprehensive assessment of music–dance alignment beyond first-order beat synchronization.
>
> We extend our original beat-level metrics with downbeat-aware counterparts: we extract audio downbeats using a standard pipeline (RNN + DBN tracking), and motion downbeats by selecting phase-consistent high-energy motion beats based on velocity signals. Specifically, we extract the audio downbeat set \( D \) and motion downbeat set \( N \), and define:
> - **DBCS**: proximity between each motion downbeat and its nearest audio downbeat;
> - **ADHS**: proportion of audio downbeats matched by motion downbeats.
>
> We define **DownbeatAlign = (DBCS + ADHS) / 2**.
> We conduct additional experiments on an auxiliary evaluation set (same protocol for all methods), with results:
>
> | Method     | DBCS ↑ | ADHS ↑ | DownbeatAlign ↑ |
> |------------|--------|--------|-----------------|
> | Veo3       | 0.20   | 0.07   | 0.14            |
> | Sora2      | 0.16   | 0.04   | 0.10            |
> | JavisDiT   | 0.08   | 0.06   | 0.07            |
> | Wan-Foley  | 0.16   | 0.06   | 0.11            |
> | Universe   | 0.17   | 0.07   | 0.12            |
> | RhyJAM     | 0.16   | 0.07   | 0.12            |
>
> These results show that incorporating downbeat-level evaluation provides a stricter and more structured assessment, while preserving consistent trends across methods. We will include this analysis in the revised version to further strengthen our evaluation framework.
>
> ## **Q4: About Professional "Music-to-Dance" Deneration Models**
> We thank the reviewer for this suggestion. We agree that comparing with specialized music-to-dance models would be valuable. However, in our investigation, we found that existing music-to-dance methods are not directly compatible with our setting for two main reasons: (1) many recent approaches are not publicly available, making reproducible comparison infeasible; (2) a large portion of prior work focuses on 3D human motion generation rather than full video generation, which differs significantly from our task formulation.
>
> Given these constraints, we adopt general audio-to-video generation models as substitutes to provide a fair and reproducible comparison.
>
> In addition, for the cascaded text-to-music-to-dance pipeline, we use Wan2.5 as the music-to-dance component. Wan2.5 is a strong video generation system that naturally supports audio-conditioned video synthesis, aligning well with the input–output format of this task. Its strong generative capability allows it to serve as a competitive proxy for music-to-dance generation in the absence of directly comparable open models.
>
> We will clarify this design choice in the revised version.

---

> > ### Author Rebuttal · Reviewer_qjM4 · 2026-04-03
> >
> > Thank you for the authors’ response. Although I still believe that it would be valuable to include comparisons with prior works specifically developed for dance generation, there are several publicly available methods that could serve as relevant baselines, such as: https://github.com/fancongyi/Danceba

---

> > > ### Author Response · Authors · 2026-04-04
> > >
> > > We thank the reviewer for the helpful follow-up and for suggesting relevant prior work such as Danceba.
> > >
> > > We agree that including specialized music-to-dance models would be valuable. However, after careful examination, we found that existing publicly available methods are not directly compatible with our cascaded setting due to the following reasons:
> > >
> > > 1. **Mismatch in conditioning interface within the cascade pipeline.**
> > >    Our cascaded pipeline follows a *text → music → (music + video prompt) → video* formulation, where the second stage requires **joint conditioning on both music and a video prompt** (e.g., scene, subject, and style).
> > >    In contrast, music-to-dance methods like Danceba, are designed for *music → dance* generation and **do not support additional video/text conditioning inputs**. Therefore, they cannot be directly integrated into our cascade without modifying their core design, which would compromise the fairness of comparison.
> > >
> > > 2. **Limited availability of open-domain inference.**
> > >   According to the authors of Danceba, the released code primarily supports evaluation on the AIST++ dataset and does not provide a ready-to-use inference pipeline for open-domain music inputs. The authors also note that the model does not currently support in-the-wild music inference and that performance in such settings is not satisfactory due to limited dataset coverage. This makes it difficult to obtain meaningful and comparable results under our open-domain benchmark setting.
> > >
> > > Given these considerations, we adopt strong general audio-to-video models as practical proxies within the cascaded framework to ensure fair and reproducible evaluation under a unified, prompt-driven setting.
> > >
> > > We will further clarify these points in the revised version. We also agree that incorporating specialized music-to-dance models is an important direction, and we will include such comparisons once models with compatible conditioning interfaces and open-domain inference support become available.

---

### Official Review · Reviewer_xr9S · 2026-03-09

**Soundness:** 2
**Presentation:** 2
**Significance:** 2
**Originality:** 2
**Overall Recommendation:** 4
**Confidence:** 3

**Summary:**

This paper aims to properly evaluate music-dance generation systems. They argue that existing audio-visual benchmarks fail to focus on beat alignment between video and audio. The paper proposes TMD-Bencg for text-to-music-and-dance generation systems, which evaluates unimodal generation quality, instruction adherence, and cross-modal rhythmic alignment. Additionally. they introduced their model for joint music-dance generation and 10k rhythm-aligned dataset.

**Compliance With Llm Reviewing Policy:**

Affirmed.

**Final Justification:**

My concerns have largely been addressed, and I would be willing to raise my score.

**Key Questions For Authors:**

1. What beat tracking method was used?
2. Will the TMD-Bench evaluation code and dataset be publicly released?
3. Since the attention map shown in Figure 5 seems to be completely diagonal, the attention map would be approximately an identical matrix. Do you think that the cross-attention mechanism can be replaced with adaLN?

**Limitations:**

yes

**Strengths And Weaknesses:**

# Strengths
1. The paper proposed an important issue for modern video-audio co-generation systems, and introduced methods to evaluate.
2. The evaluation of existing models is extensive.
# Weaknesses
1. No demo provided.
2. Music beat and singing beat detection have been studied for a long time in the area of music information retrieval. For beat detection, the paper only provides a high-level idea, which concerns me about the actual quality. For example, video beat detection should reference [1], music beat detection should reference [2].
3. No evaluation of the dataset or the proposed metrics (i.e., VBCS and ABHS). The author should use the proposed metrics to evaluate the proposed dataset and other audio-visual benchmarks. Also, human evaluation should be added to demonstrate the superior audio-visual alignment of the proposed dataset and the reliability of the proposed metrics (including low-level metrics and high-level Judgements).
4. Considering only the beat for music might not be sufficient; downbeat and the beat positions (e.g., syncopation) are also important for dance and music relations.



[1] Davis, A., & Agrawala, M. (2018). Visual rhythm and beat. ACM Transactions on Graphics (TOG), 37(4), 1-11.
[2] Böck, S., Korzeniowski, F., Schlüter, J., Krebs, F., & Widmer, G. (2016, October). Madmom: A new python audio and music signal processing library. In Proceedings of the 24th ACM international conference on Multimedia (pp. 1174-1178).

---

> ### Author Rebuttal · Authors · 2026-03-31
>
> ## **Q1: About Demo Page**
> We thank the reviewer for the suggestion. Qualitative examples are provided in the appendix (Figures 11-13), and additional playable demos are available at: https://anonymous-2312.github.io/TMD-Bench/.
>
> ## **Q2: About Detection Method Reference**
> We thank the reviewer for the helpful suggestion. Our implementation uses standard beat extraction pipelines: audio beats are obtained with`librosa.beat.beat_track`, following classical MIR practice and related systems such as Madmom [1], while video beats are derived from pose-based motion dynamics using MMPose, which is conceptually aligned with prior visual beat detection work [2]. In the revision, we will explicitly cite these references and better position our design within the MIR and visual rhythm literature.
>
> [1] madmom: a new Python Audio and Music Signal Processing Library
>
> [2] Visual Rhythm and Beat
>
> ## **Q3: About Evaluation of the Proposed Metrics**
> We thank the reviewer for this important suggestion. We have added additional experiments in the appendix by applying our MDAlign metrics to multiple data sources. As shown below,`VBCS`and`ABHS`increase from model-generated samples to random dataset samples and further to manually curated high-alignment samples, indicating that the proposed metrics track perceptual rhythm alignment and that our dataset exhibits strong rhythm-motion coupling.
> | Data source | VBCS | ABHS | CSD | HSD |
> |:---|---:|---:|---:|---:|
> | Model-generated results (RhyJAM) | 0.50 | 0.27 | 0.19 | 0.12 |
> | Random 1k subset from our 10k dataset | 0.55 | 0.35 | 0.14 | 0.13 |
> | 30 manually selected high-alignment samples | 0.62 | 0.41 | 0.17 | 0.15 |
> | Full 10k dataset | 0.53 | 0.27 | 0.23 | 0.13 |
>
> We further conducted human evaluation to assess metric reliability. On 100 generated samples rated by 10 annotators, Gemini 3.0 Pro shows strong agreement with human judgments, achieving macro-average` PLCC / SRCC / QWK = 0.6117 / 0.5916 / 0.3045`, and strong stability across 50 repeated runs (11 of 12 dimensions score 1.0). In addition, we performed a lightweight human study in which 4 annotators independently rated 100 randomly sampled data points on a 0-4 scale; the results further support the strong music-dance alignment quality of our dataset.
>
> | Annotator | Score |
> |:---|---:|
> | Annotator 1 | 3.02 |
> | Annotator 2 | 3.34 |
> | Annotator 3 | 3.18 |
> | Annotator 4 | 2.98 |
> | Avg. | 3.13 |
>
> ## **Q4: About Additional Downbeat Metrics**
> We agree that realistic music-dance coupling also depends on higher-level rhythmic structure such as downbeats. To address this, we extend our original beat-level metrics with downbeat-aware counterparts. Audio downbeats are extracted using a standard RNN+DBN pipeline, while motion downbeats are defined as phase-consistent high-energy motion beats derived from velocity signals. We define` DBCS`to measure the proximity between motion and audio downbeats, and`ADHS`to measure the proportion of audio downbeats matched by motion downbeats. We further define`DownbeatAlign`as the average of DBCS and ADHS.
> | Method | DBCS | ADHS | DownbeatAlign |
> |:---|---:|---:|---:|
> | Veo 3 | 0.20 | 0.07 | 0.14 |
> | Sora 2 | 0.16 | 0.04 | 0.10 |
> | JavisDiT | 0.08 | 0.06 | 0.07 |
> | Wan-Foley | 0.16 | 0.06 | 0.11 |
> | Universe | 0.17 | 0.07 | 0.12 |
> | RhyJAM | 0.16 | 0.07 | 0.12 |
>
> ## **Q5: About Beat Tracking Method**
> We use librosa.beat.beat_track to estimate beat timestamps from the audio track. The extracted beat timestamps are then mapped onto the clip timeline and used for MDAlign computation.
>
> ## **Q6: About Release Plan**
> We will release the code, model weights, and dataset upon acceptance.
>
> ## **Q7: About AdaLN Ablation Experiment**
> We thank the reviewer for this insightful question. The near-diagonal pattern in Figure 5 does not imply an identity mapping; it reflects monotonic alignment between temporally ordered but heterogeneous token sequences, whose bandwidth and local deviations encode cross-modal structure such as rhythmic delays, tempo variation, and one-to-many or many-to-one correspondences. This is fundamentally different from adaLN, which provides only feature-wise conditioning and cannot explicitly model token-level alignment. To verify this, we replace cross-attention with adaLN-style conditioning.
> | Method | VBCS | ABHS | Align(Gemini) | Video Avg. | Audio Avg. |
> |:---|---:|---:|---:|---:|---:|
> | RhyJAM (full) | 0.50 | 0.27 | 0.79 | 0.71 | 0.52 |
> | w/o XAttn, use adaLN | 0.59 | 0.24 | 0.73 | 0.68 | 0.55 |
>
> While VBCS increases from 0.50 to 0.59, both ABHS and perceptual alignment drop (`0.27 -> 0.24`,` 0.79 -> 0.73`), indicating weaker fine-grained temporal alignment and cross-modal consistency, with unimodal quality remaining comparable.

---

> > ### Author Rebuttal · Reviewer_xr9S · 2026-04-03
> >
> > Thank you for your response. My concerns have largely been addressed

---

### Official Review · Reviewer_zGsB · 2026-03-10

**Soundness:** 3
**Presentation:** 3
**Significance:** 2
**Originality:** 3
**Overall Recommendation:** 4
**Confidence:** 3

**Summary:**

This paper introduces TMD-Bench, a benchmark for text-driven music–dance co-generation, together with a unified baseline model named RhyJAM. The benchmark evaluates systems along multiple dimensions, including unimodal generation quality, instruction adherence, and cross-modal rhythmic alignment between music and dance. The authors also construct a rhythm-aligned dataset of approximately 10k music–dance pairs and propose evaluation metrics that combine algorithmic measures and MLLM-based perceptual judgments.

**Compliance With Llm Reviewing Policy:**

Affirmed.

**Final Justification:**

My concern is addressed. And I keep my score of 4.

**Key Questions For Authors:**

No.

**Limitations:**

I am not very familiar with the motion generation in general, but I feel the pipeline and overall framework in the paper looks good to me. However, the only criticism is that the paper does not provide some demo or visuals to illustrate the benchmark.

**Strengths And Weaknesses:**

Strengths:
1. Addresses an important and relatively underexplored problem: evaluation of music–dance co-generation.
2. Proposes a comprehensive benchmark and evaluation framework combining physical metrics and perceptual judgments.
3. Provides a curated dataset and baseline model, which could be useful resources for future research.

Weaknesses:
1. The paper lacks qualitative demo results or visual demonstrations, which would help readers better assess motion and rhythmic alignment.
2. The evaluation pipeline is somewhat complex, and it may benefit from clearer guidance on the most important metrics.

---

> ### Author Rebuttal · Authors · 2026-03-31
>
> Thanks for your recognition of our work.
> ## **Q1: About Demo Page**
> We thank the reviewer for this suggestion. We agree that qualitative demonstrations are important for assessing motion quality and rhythmic alignment. We have provided a comprehensive demo page showcasing representative results of our method, including synchronized music–dance generation examples, which can be accessed at https://anonymous-2312.github.io/TMD-Bench/.
> ## **Q2: About Clearer Guidance on Evaluation Pipeline**
> We thank the reviewer for this helpful suggestion.
>
> Our intention is to provide a comprehensive and decomposable evaluation framework covering unimodal quality, semantic adherence, and cross-modal alignment. Among these, we would like to clarify that cross-modal rhythmic alignment is the primary focus of this task, and metrics in Avalign (e.g., VBCS, ABHS, and perceptual alignment) should be considered the most critical indicators.
>
> The unimodal (audio/video) and semantic metrics serve as complementary diagnostics, helping to interpret model behavior and identify failure modes, rather than being the sole criteria for comparison. In the revised version, we will explicitly highlight the hierarchy of metrics (primary vs. auxiliary) and provide clearer guidance in the evaluation section.

---

> > ### Author Rebuttal · Reviewer_zGsB · 2026-04-05
> >
> > Thanks for providing the demo. However, the layout is a bit confusing to me. What is the usage of the music captioner? And the diverse example is the generated one or the real videos?

---

> > > ### Author Response · Authors · 2026-04-06
> > >
> > > Thank you very much for your valuable comments.
> > >
> > > Regarding the Music Captioner mentioned in the paper, we hereby further clarify its design purposes:
> > >
> > > 1. To evaluate the quality and rationality of generated music: the model analyzes a piece of music from six dimensions to determine whether it conforms to the given text prompt.
> > >
> > > 2. To annotate training data for the text-to-music generation task: we train the model based on Qwen-Omni, enabling it to describe music from multiple dimensions and output a natural language summary. This summary serves as the text prompt label of the music for the training of subsequent models.
> > >
> > > In addition, with respect to the examples on the demonstration page that may cause misunderstanding, we clarify that these content are model-generated results rather than real manually annotated data. We fully understand the potential confusion this may cause, and we solemnly promise to clearly mark such cases in the final version of the manuscript to avoid misunderstanding among readers.

---

### Official Review · Reviewer_BJFA · 2026-03-13

**Soundness:** 3
**Presentation:** 4
**Significance:** 4
**Originality:** 3
**Overall Recommendation:** 4
**Confidence:** 4

**Summary:**

This paper addresses the evaluation bottleneck in music–dance co-generation by introducing TMD-Bench, a comprehensive multi-level benchmarking framework designed to assess the intrinsic coupling between auditory rhythm and choreographic motion. The authors propose a two-level evaluation paradigm: a low-level physical track that utilizes signal-based statistics to measure objective beat-level synchronization, and a high-level perceptual track that leverages MLLM-as-a-Judge to assess semantic instruction adherence and holistic aesthetic quality. Additionally, the work presents RhyJAM, a unified flow-matching-based diffusion model trained on a newly curated 10k-scale rhythm-aligned dataset. By shifting the focus from generic audio–visual consistency to fine-grained rhythmic coherence, this work provides a structured, scalable, and reproducible methodology for benchmarking generative models in the music–dance domain.

**Compliance With Llm Reviewing Policy:**

Affirmed.

**Key Questions For Authors:**

Good paper, please see Weaknesses

**Limitations:**

The authors provide a brief impact statement; however, the discussion of the technical limitations of their work remains inadequate.
I encourage the authors to expand this discussion to include: (1) Scope and Generalization, specifically addressing the potential bias of the rhythmic alignment metrics toward percussive dance styles over fluid, non-rhythmic movement; (2) Data Bias, While the 10k dataset is a strength, the authors should discuss potential demographic or cultural biases inherent in the collection of dance videos from online platforms, and how this might influence the fairness of the benchmark.

**Strengths And Weaknesses:**

Strengths

1. The paper effectively identifies a crucial, under-addressed bottleneck in joint audio–visual generation: the fundamental mismatch between generic audio-visual consistency metrics and the fine-grained rhythmic coupling required for music–dance co-generation. This focus on "rhythmic and kinetic coherence" is timely and highly relevant to the community.

2. The paper is well-written and logically organized. The two-level evaluation paradigm (low-level physical vs. high-level perceptual) is presented clearly and provides a coherent framework that is easy for the reader to navigate.

3. The extensive effort to compile a 10k-scale rhythm-aligned dataset and the wide range of comparative experiments (across closed-source, cascaded, and open-source models) demonstrate a commendable level of empirical thoroughness.

Weaknesses

1. The paper lacks transparency regarding the training of RhyJAM. It is unclear whether the model is trained from scratch or if it initializes from a pre-trained foundation model. Given the scale of the task and the relatively short training duration (10 epochs), the absence of details regarding backbone initialization significantly obscures the claim of architectural novelty.

2. The authors rely heavily on CLAP similarity for instruction adherence. However, CLAP is known to be relatively coarse-grained. The authors fail to justify why CLAP is sufficient to capture complex, composition-specific text instructions in the music domain, potentially leading to inflated or misleading semantic compliance scores.

3. While the authors state that the Music Captioner is fine-tuned from Qwen-Omni, they omit critical implementation details. Without a clear description of the fine-tuning objectives, hyper-parameters, and the specific composition of the instruction-following data used, the reproducibility of this core component is severely compromised.

4. The choice of "Wan 2.2 + Hunyuan-Foley" as a cascaded baseline is questionable. Hunyuan-Foley is primarily designed for ambient foley sound generation, not structured music. Using a foley-specific model to evaluate music-dance alignment seems mismatched; the authors should explain why models better suited for the music domain (e.g., AudioX [1] or similar music-centric generators) were not selected for comparison.

5. The evaluation of audio generation is overly reliant on aesthetic/perceptual scores. Standard, widely-accepted quantitative metrics such as Frechet Audio Distance (FAD) or Inception Score (IS) are missing.

6. In Section 3.3, the authors mention that MDAlign uses an "MLLM-based perceptual alignment," but fail to specify the underlying model for this specific task or provide a rigorous demonstration of its reliability in measuring rhythmic sync beyond the general validation of the judge in the appendix.

7. The literature review focuses exclusively on unidirectional music-to-motion synthesis and general audio–video generation. Given the paper’s framing as a "co-generation" task, it is essential to include related works on video-driven music synthesis.

Minor: Figure 1 is not clear



[1] AudioX: A Unified Framework for Anything-to-Audio Generation

---

> ### Author Rebuttal · Authors · 2026-03-31
>
> Thanks for your recognition of our work. Due to the response length limit, we address multiple related concerns within a single response for clarity and completeness.
> ## **Q1: About RhyJAM & Music Captioner Training.**
>
> **RhyJAM training.**
> RhyJAM adopts a multi-stage training strategy with partial pretrained initialization. We first train a unified audio VAE using audio–text and audio–video–text data to obtain stable latent representations. The video branch is initialized from a pretrained Wan model, while the audio branch is trained on the learned latent space. We then perform joint audio–video training with a curriculum: (i) structured speaking scenarios for stable alignment, followed by (ii) open-domain data. Finally, we incorporate a curated beat-aligned dance dataset to further enhance synchronization.
>
> **Music Captioner.**
> We fine-tune the model (initialized from Qwen-Omni) on ~1M music–text pairs, where annotations are generated by Gemini 2.5 Pro with both structured labels and captions across six dimensions. Audio is preprocessed via UVR5, denoising, and SNR filtering. We use SFT to jointly learn structured attributes and natural language captions.
>
> **Training configuration.**
> 64×H20 GPUs, batch size 64, max length 6340, DeepSpeed FusedAdam, lr=5e-5 (cosine), warmup=1e-3, weight decay=0.0, betas=(0.9, 0.95), ε=1e-8 (full details will be included in the revision).
>
> ## **Q2: About Concerns on CLAP & MLLM-based alignment.**
>
> **Role of CLAP.**
> We use a music-specialized LAION-CLAP pretrained on large-scale music datasets, which is better aligned with music-domain audio–text correspondence. Importantly, CLAP is not the primary signal for instruction adherence, but only a low-level complementary similarity measure. To overcome its coarse granularity, our framework introduces the Music Captioner, which decomposes music into six fine-grained semantic dimensions. Instruction adherence is then evaluated via dimension-wise matching, enabling compositional and interpretable assessment beyond CLAP.
>
> **MLLM-based perceptual alignment.**
> We clarify that the evaluator used in MDAlign is Gemini 3.0 Pro (Sec. 4.2). Its reliability is validated in the appendix from two aspects:
>
> - **Accuracy**: high correlation with human judgments (PLCC/SRCC/QWK = 0.61 / 0.59 / 0.30);
>
> - **Stability**: near-perfect consistency (~1.0) over 50 repeated evaluations.
>
> We will revise the paper to make these points clearer.
>
> ## **Q3: About Additional Metrics and Limitations Discussion.**
>
> **Quantitative audio metrics.**
> We agree that standard metrics such as FAD and IS provide important complementary signals. We have now added FAD (CLAP, MERT embeddings) and IS (PANNs) for all baselines. These results will be included in the revised version to provide a more comprehensive evaluation.
>
> |Method| FAD (CLAP)|FAD (MERT)|IS (PANNs)|
> |-|-|-|-|
> |Veo3|0.45|11.97|1.75|
> |Sora2|0.43|8.01|1.66|
> |Seedance 1.5 Pro|0.53|17.29|1.58|
> |Kling 2.6| 0.50|14.81|1.80|
> |Wan 2.6|0.60|14.31|2.05|
> |Ace-Step|0.39|8.91|1.95|
> |Foley|0.49|17.95|1.68|
> |Universe|0.36|16.96|2.10|
> |JavisDiT|0.66|22.45|1.47|
> |Ovi|0.66|24.93|1.68|
> |Ltx-2| 0.61|21.90|1.63|
> |RhyJAM|0.69|25.66|1.49|
>
> **Limitations.**
> We will expand the discussion in the revised version:
>
> - Our rhythmic alignment metrics may favor percussive, beat-salient dances, while more fluid or expressive motions are harder to capture with event-based alignment. Although MLLM-based perceptual evaluation partially mitigates this, designing metrics that better model continuous motion dynamics remains future work.
>
> - The dataset is collected from online platforms and may contain demographic and cultural biases, with popular styles overrepresented. This could influence both training and evaluation.
>
> ## **Q4: About Cascaded Baseline Choice.**
>
> While Foley was originally included to represent a generic cascaded video–audio pipeline, we have now added a comparison by replacing Foley with AudioX.
>
> | Method  | Audio Avg. | Align (Gemini) | ABHS |VBCS|
> |-|-|-|-|-|
> | RhyJAM| 0.52| 0.79| 0.27| 0.50|
> | Wan + AudioX| 0.44| 0.65| 0.22| 0.44|
> | Wan + Foley| 0.48| 0.77| 0.25| 0.50|
>
> The overall conclusion remains unchanged: cascaded pipelines struggle to achieve strong cross-modal rhythmic synchronization. We will include these results in the revised version.
>
> ## **Q5: Video-to-Music Related Work.**
>
> We will revise the related work to cover representative V2M approaches, including early feature-mapping methods (e.g., CMT), learning-based alignment models (e.g., MuVi, VidMuse), and unified multimodal frameworks (e.g., AudioX, MuMu-LLaMA).
>
> ## **Q6: About Figure Clarity.**
> We thank the reviewer for pointing this out. The top-right image in Figure 1 is over-compressed, causing low clarity. We will replace it with a higher-resolution version and adjust export settings in the revision.

---

> > ### Author Rebuttal · Reviewer_BJFA · 2026-04-05
> >
> > Thanks for the response. My concerns have been addressed.

---

### Decision · Program_Chairs · 2026-04-30

**Decision:**

Accept (regular)

**Comment:**

All reviewers provided positive evaluations with scores of Weak Accept. They agreed that the paper addresses an interesting and timely problem and makes a solid technical contribution. The authors’ rebuttal adequately clarified the reviewers’ questions, and no critical concerns remain.

Based on the consistent reviewer support, I recommend acceptance.